# A subcellular map of the human kinome

**Haitao Zhang[1,2], Xiaolei Cao[1], Mei Tang[1], Guoxuan Zhong[1], Yuan Si[1], Haidong Li[3], Feifeng Zhu[1], Qinghua Liao[1], Liuju Li[4], Jianhui Zhao[5], Jia Feng[6], Shuaifeng Li[1], Chenliang Wang[1], Manuel Kaulich[7], Fangwei Wang[1,2], Liangyi Chen[4], Li Li[8], Zongping Xia[1], Tingbo Liang[2,5], Huasong Lu[1,2], Xin-Hua Feng[1,2], Bin Zhao[1,2,5]\***

[1]The MOE Key Laboratory of Biosystems Homeostasis & Protection, Zhejiang Provincial Key Laboratory for Cancer Molecular Cell Biology, and Innovation Center for Cell Signaling Network, Life Sciences Institute, Zhejiang University, Hangzhou, China; [2]Cancer Center, Zhejiang University, Hangzhou, China; [3]College of Biology and Pharmacy, Yulin Normal University, Yulin, China; [4]State Key Laboratory of Membrane Biology, Beijing Key Laboratory of Cardiometabolic Molecular Medicine, Institute of Molecular Medicine, Peking University, Beijing, China; [5]Department of Hepatobiliary and Pancreatic Surgery, Zhejiang Provincial Key Laboratory of Pancreatic Disease, The First Affiliated Hospital, School of Medicine, Zhejiang University, Hangzhou, China; [6]Department of Ophthalmology, The Children's Hospital, School of Medicine, and National Clinical Research Center for Child Health, Zhejiang University, Hangzhou, China; [7]Institute of Biochemistry II, Goethe University Frankfurt-Medical Faculty, University Hospital, Frankfurt, Germany; [8]Institute of Aging Research, Hangzhou Normal University, Hangzhou, China

**\*For correspondence:**
binzhao@zju.edu.cn

**Competing interests:** The authors declare that no competing interests exist.

**Abstract** The human kinome comprises 538 kinases playing essential functions by catalyzing protein phosphorylation. Annotation of subcellular distribution of the kinome greatly facilitates investigation of normal and disease mechanisms. Here, we present Kinome Atlas (KA), an image-based map of the kinome annotated to 10 cellular compartments. 456 epitope-tagged kinases, representing 85% of the human kinome, were expressed in HeLa cells and imaged by immunofluorescent microscopy under a similar condition. KA revealed kinase family-enriched subcellular localizations and discovered a collection of new kinase localizations at mitochondria, plasma membrane, extracellular space, and other structures. Furthermore, KA demonstrated the role of liquid-liquid phase separation in formation of kinase condensates. Identification of MOK as a mitochondrial kinase revealed its function in cristae dynamics, respiration, and oxidative stress response. Although limited by possible mislocalization due to overexpression or epitope tagging, this subcellular map of the kinome can be used to refine regulatory mechanisms involving protein phosphorylation.

## Introduction

The kinome, encoded by about 2% of the human genome, is one of the largest superfamilies of homologous proteins consisting of 538 kinases (*Manning et al., 2002b*). These enzymes catalyze the transfer of γ-phosphate of ATP to the alcohol groups (on Ser and Thr) or phenolic groups (on Tyr) of proteins to generate phosphate monoesters. As a consequence, the substrate protein could be altered in many ways, such as activity, subcellular localization, and interacting proteins (*Cohen, 2002*; *Manning et al., 2002a*). Therefore, protein phosphorylations function as molecular switches in essentially every aspect of cellular activities. Mutation and dysregulation of protein kinases are causal to many diseases such as cancer (*Greenman et al., 2007*; *Lahiry et al., 2010*). Kinases are also among the most important targets for precision medicine (*Klaeger et al., 2017*; *Zhang et al., 2009*).

Spatial partitioning of molecules into functional compartments is a basic principle for the organization of a cell (*Bauer et al., 2015*). Compartmentation determines the chemical environment, and thus profoundly affects kinase activity and substrate selectivity. Kinases were found in most organelles, and study of subcellular localization on the proteome level facilitated mapping of the kinome (*Thul et al., 2017*). In imaging-based approaches, antibodies were used to label proteins for visualization. For instance, Cell Atlas of the Human Protein Atlas (HPA) project mapped more than 12,000 proteins to 13 major organelles using antibodies against endogenous proteins (*Thul et al., 2017*). However, reliability was limited by antibody specificity in such one antibody one protein approaches (*Baker, 2015*). Alternatively, organelle proteome could be identified by mass spectrometry after fractionation (*Calvo et al., 2016*; *Orre et al., 2019*). Although empowered by proximity-labeling technologies (*Branon et al., 2018*; *Lönn and Landegren, 2017*), compartments accurately analyzed this way are still limited. Furthermore, curation of subcellular localization data from UniProt, literatures, and open resources has resulted in comprehensive databases such as COMPARTMENTS (*Binder et al., 2014*). However, accuracy is often affected by complex data source. As a result, understanding of subcellular localizations of the kinome is still fragmented.

Here, we report a subcellular map of the kinome generated by immunofluorescent (IF) imaging of ectopically expressed epitope-tagged kinases in HeLa cells. This dataset, the Kinome Atlas (KA), annotated 456 kinases to 10 compartments of the cell. Although mislocalization was possible due to overexpression or epitope tagging, KA technically complements previous approaches in presenting a whole picture of the kinome. Experiments done in the same cell line transfected and cultured in a similar condition allowed comparison within the dataset, revealing localization features of kinase families and a broader role of liquid-liquid phase separation (LLPS) in formation of both nuclear and cytosolic kinase condensates. This study not only discovered a collection of new kinase localizations, but also demonstrated the role of MOK in mitochondrial structure and functions. This resource lays a foundation for further investigation of kinases and protein phosphorylation in terms of regulation and functions.

## Results

### Mapping subcellular localizations of the kinome

We constructed an expression plasmid library of the kinome using gateway cloning as described in the Materials and methods section. 464 protein kinase-encoding plasmids were got (*Supplementary file 1*), representing 86.2% of the human kinome, and covering each family similarly (*Figure 1A, Figure 1—figure supplement 1A*). 95.1% of genes in the library are of human origin, and the rest are of mouse or rat origins (*Figure 1B*). All clones are epitope-tagged, with 46.8% in Flag tag, 46.5% in tandem HA/Flag tags, 3.4% in HA tag, 2.4% in Myc tag, and 0.9% in GFP tag. 54.7% of the clones have epitope tags on the amino (N)-terminal, and 45.3% of clones are tagged on the carboxyl (C)-terminal (*Figure 1B*). The protein kinase-like (PKL) family is overrepresented with N-terminal tag, while the nucleoside diphosphate kinases (NDK) and tyrosine kinases (TK) families have higher ratio of C-terminal tag (*Figure 1—figure supplement 1B*). We evaluated expression of the library by transient transfection using about half of the library. By western blotting, 37% of plasmids showed high level of expression, 58.5% had medium or low expression, and 4.5% could not be detected (*Figure 1C*).

To determine kinase subcellular localizations at the kinome level, we transfected each plasmid into HeLa cells and stained for the epitope tag (*Figure 1—figure supplement 1C*). According to Cell Atlas of the HPA project (*Thul et al., 2017*), 10 subcellular compartments were annotated, including cytosol (C), nucleus (N), plasma membrane (PM), mitochondrion (MI), endoplasmic reticulum (ER), Golgi apparatus (GL), vesicle (V), cytoskeleton (CS), centrosome (CT), and aggresome (AG). Two representative fields were captured for each kinase and the images were deposited to the Cell Image Library database (http://cellimagelibrary.org/pages/kinome_atlas). Annotation was based on patterns of kinase localization similar to that of an organelle marker protein as demonstrated in a similar condition (*Figure 1D*). A sum of 10 points were assigned to each kinase to reflect their distribution across the 10 compartments (*Supplementary file 2*). A total of 456 kinases were annotated, while the other 8 kinases were not detected. We defined specific localization to a compartment as a localization score of 5 or more. When the score was 5/5 between cytosol and another compartment,

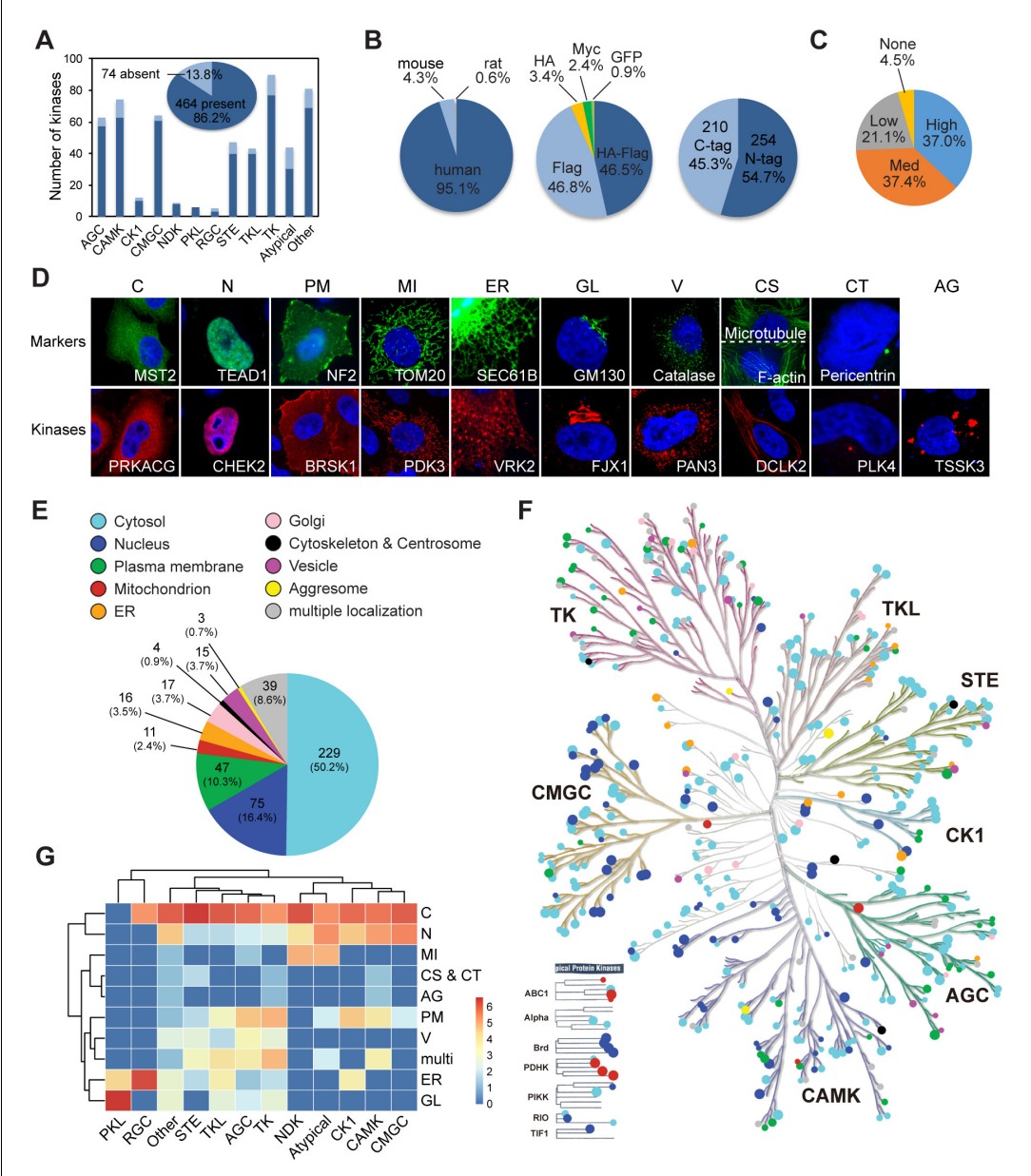

**Figure 1.** Mapping subcellular localizations of the kinome. (**A**) Coverage of the kinome (pie graph) and kinase families (bar graph) by the plasmid library. (**B**) Composition of the kinome library. Species origins (left), epitope tags (middle), and tag positions (right) were summarized. (**C**) Expression levels of kinase plasmids. HeLa cells were transfected and expression levels were determined by western blotting. (**D**) Ten cellular compartments with organelle markers (top) and representative kinases (bottom). SEC61B was visualized by GFP tag. TOM20, GM130, Catalase, microtubule, and pericentrin were stained by specific antibodies. F-actin was marked by phalloidin. Other proteins were transfected and stained for epitope tags. (**E**) Subcellular distribution of the kinome. (**F**) Kinome tree with localization information. Each dot represents a kinase. Color denotes compartment and size reflects localization score. (**G**) Family-enriched distributions for kinases. Kinase families were clustered by the ratio of localization to different compartments (*Figure 1—figure supplement 1*).

The online version of this article includes the following source data and figure supplement(s) for figure 1:

**Source data 1.** Raw data for *Figure 1* and *Figure 1—figure supplement 1*.

**Figure supplement 1.** Mapping subcellular localizations of the kinome.

it was defined specific to the non-cytosolic compartment. A kinase was called multilocalizing if its localization scores at any localization were less than 5. Representative kinases with specific localizations are shown in *Figure 1D*. This collected resource was referred to as the KA.

In the initial screen, about 50.2% of the kinome mainly localized in the cytosol, followed by 16.4% in nucleus, and 10.3% on plasma membrane (*Figure 1E*). Each of mitochondria, ER, Golgi, and vesicle structures held less than 5% of the kinome. Three kinases DCLK2, MAP4K3, and ABL2 exhibited specific localization to cytoskeleton-like structures, but 12 other kinases exhibited some level of cytoskeletal localization (*Supplementary file 2*). Only two kinases PLK4 and PLK1 had centrosomal localization. Aggresomes were distinguished from vesicles mainly by an irregular shape or aberrant distribution pattern, and were adopted only by WNK4, TSSK3, and STYK1 as specific localization (*Supplementary file 2*). Interestingly, plotting of subcellular localizations onto the kinome tree using KinMap (*Eid et al., 2017*) revealed a family-centered enrichment of cellular compartments (*Figure 1F*), which was confirmed by clustering analysis (*Figure 1G*). 77.5% of the STE (homologues of yeast Ste7, Ste11, and Ste20 kinases) family were in cytosol, significantly higher than other families (*Figure 1—figure supplement 1D*). On the other hand, CMGC (CDK, MAPK, GSK3, and CLK kinases), atypical, and calcium/calmodulin-dependent protein kinases (CAMK) families were more enriched for nuclear kinases, 39.0, 34.5, and 30.2%, respectively. The TK family was enriched for kinases on the plasma membrane, largely due to the receptor tyrosine kinase (RTK) subfamily. The AGC (family of protein kinases A, G, and C) family was also overrepresented on the plasma membrane reflecting their functions in relaying extracellular signals. Most mitochondrial kinases were in the atypical kinase family. Finally, the receptor guanylate cyclase (RGC) and PKL families were largely formed by ER- and Golgi-localized kinases. Shared localization by the same family is likely due to common evolutionary ancestor and structural similarities, and suggests related functions.

## KA complements current databases for kinome localization

We compared KA with protein subcellular localization databases for coverage of the kinome. COMPARTMENTS has four evidence channels: knowledge, experiments, text-mining, and predictions (*Binder et al., 2014*). Information from the knowledge channel is based on annotations from UniProtKB and other resources, with a confidence score from one star (lowest confidence) to five stars (highest confidence). We considered only high-confidence localizations with at least four stars, resulting in 429 kinases with an average of 2.3 localizations per kinase. Less reliable information from text-mining and predictions was not included. The experiments channel holds data from the HPA project and was thus separately compared to KA. There are 438 kinases with an average of 1.3 'main' localizations in HPA. 6 kinases were not covered by any of the three resources, and KA annotated 18 kinases on top of COMPARTMENTS and HPA (*Figure 2A* and *Supplementary file 3*). Manual literature search eliminated 12 kinases with at least one report, resulting in 6 kinases annotated for the first time, including STK32C, SBK2, STKLD1, TSSK1B, NME8, and SRPK3, which were all cytosolic in KA (*Figure 2B*).

Among 337 kinases documented by both KA and COMPARTMENTS, 71.2% matched with one of the high-confidence localizations in COMPARTMENTS (*Figure 2—figure supplement 1A*). These matched localizations were mainly cytosol, nucleus, and plasma membrane. However, 97 kinases did not match with COMPARTMENTS. 27 among them were cytosolic in KA, but mainly plasma membrane, Golgi, extracellular space, and mitochondria-localized in COMPARTMENTS (*Figure 2—figure supplement 1B*). The ratio of N-terminal tag (70.4%) in these 27 kinases was higher than that in the whole library (54.7%) suggesting that for some kinases the N-terminal tag had interfered with localizations requiring N-terminal signal peptide. However, among 187 N-tagged kinases with high-confidence localizations in COMPARTMENTS, 131 were consistent with that found in KA. Among the 56 differential kinases, 34 had membrane-related localizations in COMPARTMENTS. An additional four N-tagged kinases had transmembrane regions and predicted signal peptides. Thus, we designed C-tagged version plasmids for these 38 plasmids (*Supplementary file 1*). Following a similar imaging procedure, we found 9 kinases could not be detected and 8 kinases were not affected by tag position, but 21 kinases had different subcellular localization pattern. The KA was therefore revised (*Supplementary file 2*). To further validate specific ER, GL, CS, and CT localizations with a score $\geq 5$, we carried out co-staining with respective markers. 36 out of 42 kinases were validated, and 6 kinases were revised in *Supplementary file 2*. The kinome tree plot of kinase subcellular localizations was also revised (*Figure 2—figure supplement 1C*). Various intracellular vesicles were not further

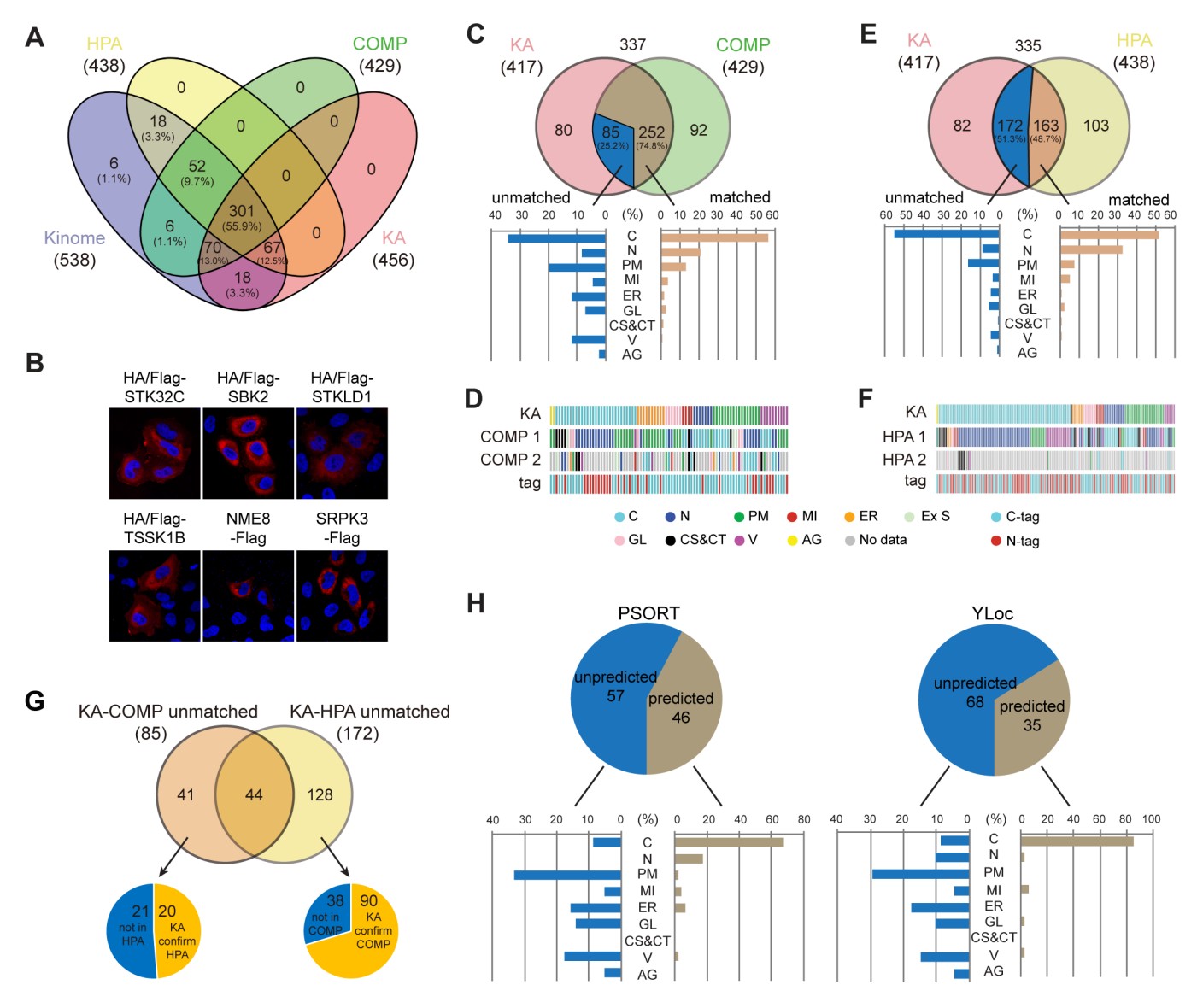

**Figure 2.** Kinome Atlas (KA) complements current databases for kinome localization. (**A**) Coverage of the kinome by COPARTMENTS, Human Protein Atlas (HPA), and KA in Venn diagram. (**B**) Subcellular localization of six kinases first annotated by KA. (**C, D**) Comparison of annotations in revised KA and high-confidence localizations in COMPARTMENTS. Distributions of matched and unmatched localizations are shown by bar graph. The heatmap demonstrates kinases (column) with color-coded localizations or position of epitope tag (row). Top two localizations in COMPARTMENTS are shown. (**E, F**) Comparison of annotations in HPA and KA. Analyses were similar to (**C, D**). (**G**) KA provides unique kinase localizations different from databases. Kinases showing different annotations in KA-COMPARTMENTS or KA-HPA comparisons were further consolidated. (**H**) Prediction of KA-unique kinase localizations. Localizations of unique kinases in (**G**) were predicted by WoLF PSORT (left panel) or YLoc (right panel). Distribution of predicted or unpredicted kinases is shown by bar graph (*Figure 2—figure supplement 1*).

The online version of this article includes the following source data and figure supplement(s) for figure 2:

**Source data 1.** Raw data for *Figure 2* and *Figure 2—figure supplement 1*.

**Figure supplement 1.** Kinome Atlas (KA) complements current databases for kinome localization.

validated by co-staining, and new MI and PM localization was validated below by co-staining and fractionation, respectively.

Comparison between revised KA and COMPARTMENTS revealed 74.8% localizations were consistent (*Figure 2C* and *Supplementary file 4*). 29 differential kinases were cytosolic in KA, among which 17 had nuclear localization in COMPARTMENTS (*Figure 2D*). It is possible that these kinases

shuttle between cytosol and the nucleus. Indeed, half of them had low levels of nuclear localization in KA. When compared with HPA, a higher ratio of kinases exhibited different localizations (*Figure 2E* and *Supplementary file 5*). An important reason was antibody specificity in HPA because

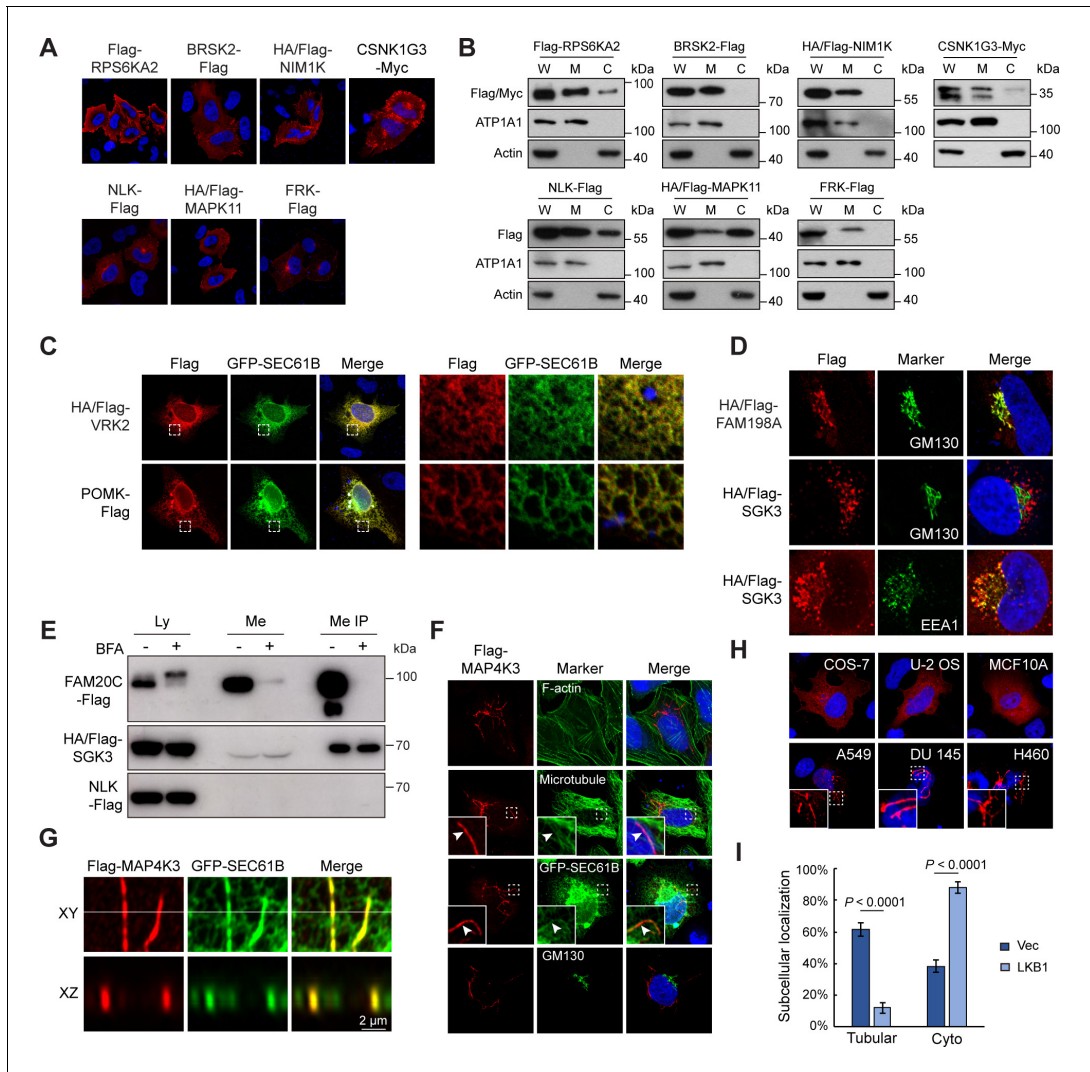

**Figure 3.** Kinome Atlas (KA) identified unknown localizations for kinases. (**A, B**) New plasma membrane-localized kinases identified by KA. Original mapping data (**A**) and confirmation by cellular fractionation (**B**). W: whole cell lysate; M: membrane; C: cytosol. Equal loading was achieved by normalizing to the same number of cells. (**C**) Confirmation of endoplasmic reticulum (ER)-localized kinases. HeLa cells were co-transfected with kinase and GFP-SEC61B. VRK2 was used as a positive control of ER protein. Squared areas were enlarged on the right. (**D**) Confirmation of SGK3 localization in transfected HeLa cells. (**E**) SGK3 was secreted to culture medium. HEK293T cells were transfected and brefeldin A (BFA) (0.5 μg/ml) treated for 24 hr as indicated. Both culture medium and cell lysates were collected. Medium was further immunoprecipitated by anti-Flag antibody. Loading was normalized by the number of input cells, and IP samples were loaded at 5× of input. (**F**) MAP4K3 colocalizes with microtubule and ER in HeLa cells. Arrowheads indicate colocalization. (**G**) MAP4K3 colocalizes with SEC61B as captured by Zeiss Airyscan high-resolution microscopy. (**H**) Cell-type-dependent localization of MAP4K3. Indicated cells were transfected with Flag-MAP4K3 and stained by anti-Flag antibody. (**I**) LKB1 inhibits tubule-like localization of MAP4K3. Control and LKB1-expressing H460 cells were transfected with Flag-MAP4K3 and were stained by anti-Flag antibody. Cells with different localization patterns were quantified. Data was represented as mean ± SD; n = 3 independent experiments, two-tailed Student's *t* test; n = 100 cells were analyzed in each experiment (*Figure 3—figure supplement 1*).

The online version of this article includes the following source data and figure supplement(s) for figure 3:

**Source data 1.** Uncropped western blot for *Figure 3*.

**Source data 2.** Raw data for *Figure 3*.

**Figure supplement 1.** Kinome Atlas (KA) identified unknown localizations for kinases.

**Figure supplement 1—source data 1.** Uncropped western blot for *Figure 3—figure supplement 1*.

if only 229 kinases with more reliable 'enhanced' or 'supported' localizations were included, the ratio of matched localizations increased by 10% (*Figure 2—figure supplement 1D*). Different from KA-COMPARTMENTS comparison, KA-HPA unmatched localizations were higher in cytosol (51.5%), and more than half of these kinases were mainly nuclear in HPA (*Figure 2F, Figure 2—figure supplement 1E*). This observation is consistent with the finding that the number of nuclear proteins in HPA Cell Atlas considerably exceeds previous reports (*Thul et al., 2017*), suggesting a tendency of nuclear background for antibodies in the HPA collection.

For some kinases, COMPARTMENTS and HPA annotated different localizations. Among them, KA confirmed 20 HPA-documented localization and 90 COMPARTMENTS-documented localization (*Figure 2G* and *Supplementary file 6*). Furthermore, KA uniquely annotated 103 kinase localizations different from other databases (*Supplementary file 6*). Importantly, these localization were largely unpredictable by main stream subcellular localization prediction tools such as WoLF PSORT and YLoc (*Briesemeister et al., 2010*; *Horton et al., 2007*; *Figure 2H* and *Supplementary file 6*), suggesting new localization mechanisms. Taken together, KA complements current databases for subcellular localization of the kinome in the same cell line transfected and cultured in a similar condition.

## KA identified a collection of unknown kinase localizations

We further screened the 103 kinase localizations uniquely annotated by KA. 58 kinases were subtracted for any previous reports escaped database curation. Selected kinases from the remaining 45 were further validated. Seven kinases exhibited novel plasma membrane localization in KA (*Figure 3A*). Cellular fraction confirmed that RPS6KA2, BRSK2, NIM1K, CSNK1G3, and NLK were mainly membranous, while MAPK11 and FRK were partially in membrane fractions (*Figure 3B*). POMK has an ER-related function in phosphorylating glycosylation-specific *O*-mannose and is mutated in a neuromuscular disorder dystroglycanopathies (*Di Costanzo et al., 2014*; *Yoshida-Moriguchi et al., 2013*). By co-expression with GFP-tagged SEC61B, an ER resident protein, we confirmed the ER localization of POMK (*Figure 3C*). SGK3 was enriched in vesicles around the Golgi, with lower expression in the Golgi itself (*Figure 3D*). In addition, SGK3 colocalized with endosomal marker EEA1 as previously reported (*Xu et al., 2001*; *Figure 3D*, *Figure 3—figure supplement 1A*). We speculated that SGK3 and other Golgi-localized kinases might also be secreted to the extracellular space. Cell lysates and culture mediums were collected from transfected cells, and were further concentrated by immunoprecipitation. FAM20C but not NLK could readily be detected in medium before and after immunoprecipitation, serving as positive and negative controls (*Zhang et al., 2018*; *Figure 3E*). Interestingly, SGK3, but not other kinases examined, could be immunoprecipitated from culture medium (*Figure 3E, Figure 3—figure supplement 1B*), indicating it was secreted, although at a much lower level than FAM20C. Furthermore, C-terminal-tagged SGK3 was also secreted (*Figure 3—figure supplement 1C*). SGK3 lacks a signal peptide, but instead has a PX domain, suggesting it may be secreted unconventionally in exosomes, which was supported by two experiments. First, treatment of cells with brefeldin A (BFA), an inhibitor of ER/Golgi-trafficking (*Misumi et al., 1986*), blocked secretion of FAM20C, but not SGK3 (*Figure 3E*). Second, when exosomes were sedimented from medium by ultracentrifugation, SGK3 but not FAM20C was enriched in the pellet fraction (*Figure 3—figure supplement 1D*). Thus, SGK3 is a new exosomal kinase, which may mediate cellular communication with the microenvironment.

MAP4K3 has been shown regulated by amino acids and play a role in nutrient regulation of mTOR signaling (*Yan et al., 2010*). We found MAP4K3 in tubule-like structures in KA. Co-staining indicated that MAP4K3 structures were colocalizing with a small fraction of microtubules (*Figure 3F*). These tubules were also non-overlapping with lysosomes even in amino acid starvation condition or in cells treated with vacuolin-1, a chemical that cause lysosome fusion and enlargement (*Figure 3—figure supplement 1E*). However, MAP4K3 tubules were overlapping with ER (*Figure 3F*), which was confirmed by super-resolution microscopy (*Figure 3G*). Interestingly, MAP4K3 tubules were also obvious in A549, Du 145, and H460 cells, all of which, as well as HeLa cells, have mutation of the *LKB1* gene, an upstream regulator of mTOR (*Corradetti et al., 2004*; *Figure 3H*). Surprisingly, MAP4K3 were mainly cytoplasmic with some small puncta in LKB1-wildtype COS-7, U-2 OS, and MCF10A cells. Furthermore, restoration of LKB1 expression in H460 cells markedly inhibited MAP4K3 tubules (*Figure 3I*). We noticed that the MAP4K3 cDNA in the library was the same as originally reported (*Diener et al., 1997*), which lacks N-terminal 12 amino acids compared with reference sequence. We therefore cloned the full-length MAP4K3 with N- and

C-terminal tags, and confirmed their tubule-like localization in HeLa cells (*Figure 3—figure supplement 1F, G*). Taken together, KA identified many kinases with previous unknown subcellular localizations, suggesting new functions and regulatory mechanisms.

We further examined the impact of kinase activity on localization for a few kinases with specific localizations. Seven kinases with PM, MI, tubule-like, and puncta patterns were inactivated by point mutations. Except HIPK2, all other inactive kinases showed localization patterns similar to their wild-type counterparts (*Figure 3—figure supplement 1H*). While wildtype HIPK2 were mostly in nuclear puncta, inactive HIPK2 was diffusive in cytoplasm in more than 50% of cells. Thus, subcellular localization of kinases could be regulated by their activity and should be systematically investigated in the future.

## Phase separation assembles kinase condensates

Distinct sub-nuclear patterns were observed for kinases. For instance, TRIB1, CDK10, CDK7, HASPIN, and STK17A were evenly distributed in nucleoplasm, while VRK3, BRD3, CDK8, CDKL3, TRIM24, and RPS6KA4 were enriched in periphery regions of the nucleus (*Figure 4A*). By high-resolution microscopy, all six nuclear peripheral kinases were found within the nuclear boundary (*Figure 4—figure supplement 1A*). Interestingly, it was reported that VRK3 phosphorylated nuclear envelope protein barrier-to-autointegration factor (BAF), thus played a role in cell cycle regulation (*Park et al., 2015*). Other kinases enriched in nuclear periphery regions may therefore also have sub-compartment-specific functions. Two kinases CDK13 and STK19 were found in structures reminiscent of nucleoli (*Figure 4A*), which was confirmed by co-staining with a nucleolar protein UTP25 (*Figure 4—figure supplement 1B*), suggesting their potential roles in ribosome biogenesis.

Remarkably, we also noticed several nuclear kinases unevenly distributed in puncta of varying size or even in bubble-like structures (*Figure 4A*). Furthermore, similar structures were also found in cytosol (*Figure 4—figure supplement 1C*). Puncta were more abundant in cells with high expression levels, suggesting concentration-dependent LLPS as an underlying mechanism. Protein kinases such as DYRK1A were reported to function in structures formed by LLPS (*Lu et al., 2018*). We tested the role of LLPS in formation of kinase condensates first using 1,6-hexanediol (Hex), a compound perturbing weak hydrophobic interactions to disassemble structures that exhibit liquid-like properties. Nuclear puncta formed by TRIB3 and HIPK2, as well as cytoplasmic puncta of BMP2K and CIT were largely eliminated by Hex (*Figure 4B*). However, Hex did not eliminate puncta formed by other kinases examined (*Figure 4—figure supplement 1D*). Although structures formed by LLPS may not be sensitive to Hex treatment due to possibilities such as liquid-to-solid transition, we chose Hex-sensitive structures for further analysis to avoid complex interpretations. We confirmed similar localization pattern using GFP-tagged kinases (*Figure 4—figure supplement 1E*). Time-lapse live-cell imaging revealed frequent puncta movements and fusion events for TRIB3, HIPK2, and BMP2K, demonstrating their dynamic and liquid-like properties (*Figure 4C*). LLPS of proteins are mainly mediated by multivalent interactions through multiple folded domains or intrinsically disordered regions (IDRs) (*Boeynaems et al., 2018*). All three kinases lack multiple protein-protein interaction domains, but have IDRs predicted by IUPred2A and PLAAC (*Figure 4D*). However, IDRs were also frequently predicted when analyzing randomly selected five nuclear kinases and five cytosolic kinases with even distribution pattern (*Figure 4—figure supplement 1F, G*), suggesting that predicted IDR is not sufficient for LLPS. Importantly, among all kinases analyzed, HIPK2, and BMP2K uniquely have prion-like regions overlapping with IDRs, which are largely devoid of charged residues, but enriched for polar residues, and are frequently involved in LLPS (*Figure 4D*, *Figure 4—figure supplement 1F, G*). Furthermore, deletion of IDRs eliminated puncta of TRIB3 and BMP2K, supporting IDR-mediated LLPS (*Figure 4E*). However, deletion of both IDRs in HIPK2 interrupted nuclear but not puncta localization, suggesting the involvement of other unpredicted region. In addition, the IDR of BMP2K, but not other kinases, formed puncta similar to the full-length protein (*Figure 4E, Figure 4—figure supplement 1H*). To determine whether BMP2K could form puncta through LLPS on endogenous expression level, we generated BMP2K 3xHA tag knock-in HeLa cells through CRISPR/Cas9-mediated recombination (*Figure 4—figure supplement 1I*). Successful knock-in was confirmed by western blotting (*Figure 4—figure supplement 1J*). Two types of BMP2K patterns were found, in cytoplasmic puncta or both in cytoplasmic puncta and nuclei (*Figure 4F*). Nuclear BMP2K was not observed in overexpression expression experiments and could be due to a different isoform. Importantly, cytoplasmic puncta of endogenous BMP2K were also eliminated by Hex treatment

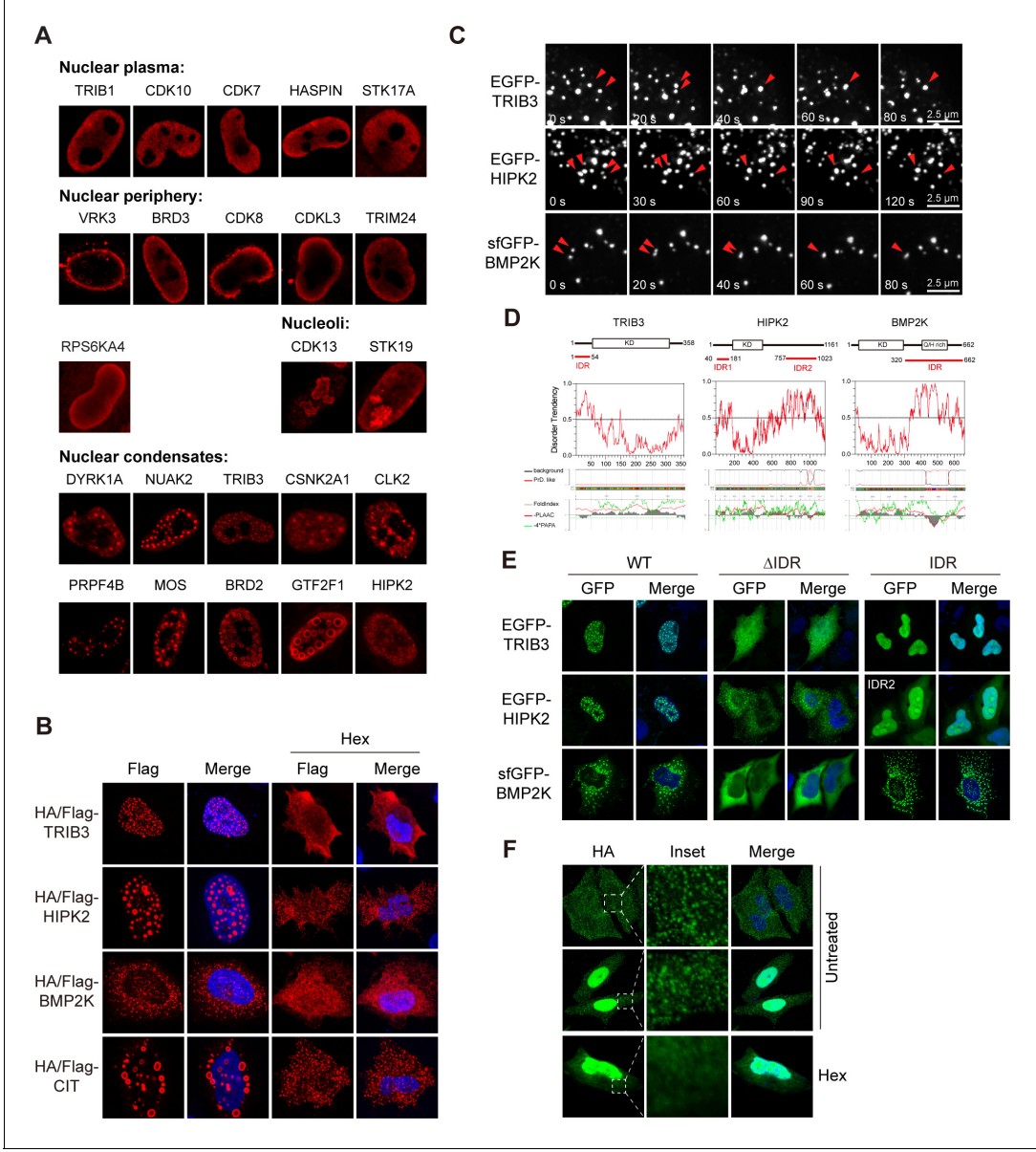

**Figure 4.** Phase separation assembles kinase condensates. (**A**) Sub-nuclear patterns of kinases in Kinome Atlas (KA). (**B**) Hex disrupts puncta localization of kinases. Transfected HeLa cells were treated with 10% 1,6-hexanediol (Hex) for 1 min as indicated. Cells were then fixed and stained. (**C**) Fusion of kinase puncta in live cells. HeLa cells expressing GFP-tagged kinases were examined by time-lapse microscopy. Fusion events were indicated by arrowheads. (**D**) Sequence analysis identified intrinsically disordered regions (IDRs) in kinases. Domain organizations were illustrated in scale. KD: kinase domain. Disorder tendency was predicted by IUPred2A. Prion-like regions and fold index were predicted by PLAAC. (**E**) Deletion of IDR impairs puncta localization. Wildtype or mutants with GFP tag were expressed and visualized in HeLa cells. For HIPK2, both IDRs were deleted in ΔIDR. (**F**) Puncta formed by endogenous BMP2K in HeLa cells. BMP2K 3xHA tag knock-in HeLa cells were treated with 10% Hex for 1 min as indicated. Cells were then fixed and stained (*Figure 4—figure supplement 1*).

The online version of this article includes the following source data and figure supplement(s) for figure 4:

**Figure supplement 1.** Phase separation assembles kinase condensates.

**Figure supplement 1—source data 1.** Uncropped western blot for *Figure 4—figure supplement 1*.

(*Figure 4F*). Taken together, LLPS may have a broader role in kinase subcellular localization and functions.

## KA identified novel mitochondrial kinases

Mitochondria generate ATP and play multiple regulatory roles in cells. KA recorded 22 kinases with some level of mitochondrial localization (*Figure 5A* and *Supplementary file 2*). 14 of them have been documented in mitochondrial proteome database MitoCarta2.0 (*Calvo et al., 2016*; *Figure 5A*), among which 6 were confirmed by co-staining with mitochondrial marker translocase of outer membrane 20 (TOM20) (*Figure 5—figure supplement 1A*). ADCK5 and OBSCN were absent from the library, and mitochondrial localization of FASTK and PAK5 has not been observed in KA. We confirmed that FASTK and PAK5 tagged on either end were non-mitochondrial (*Figure 5B*). It was reported that FASTK transcribed from an alternative start codon missing N-terminal 34 amino acids was mitochondrial (*Jourdain et al., 2015*), which we could confirm (*Figure 5—figure supplement 1B*). The functional significance of these two isoforms worth further investigation. Remarkably, we found eight new mitochondrial kinases, which were confirmed by co-staining with TOM20 (*Figure 5A, C*, *Figure 5—figure supplement 1C*). The mitochondrial localization of MOK and TNK1

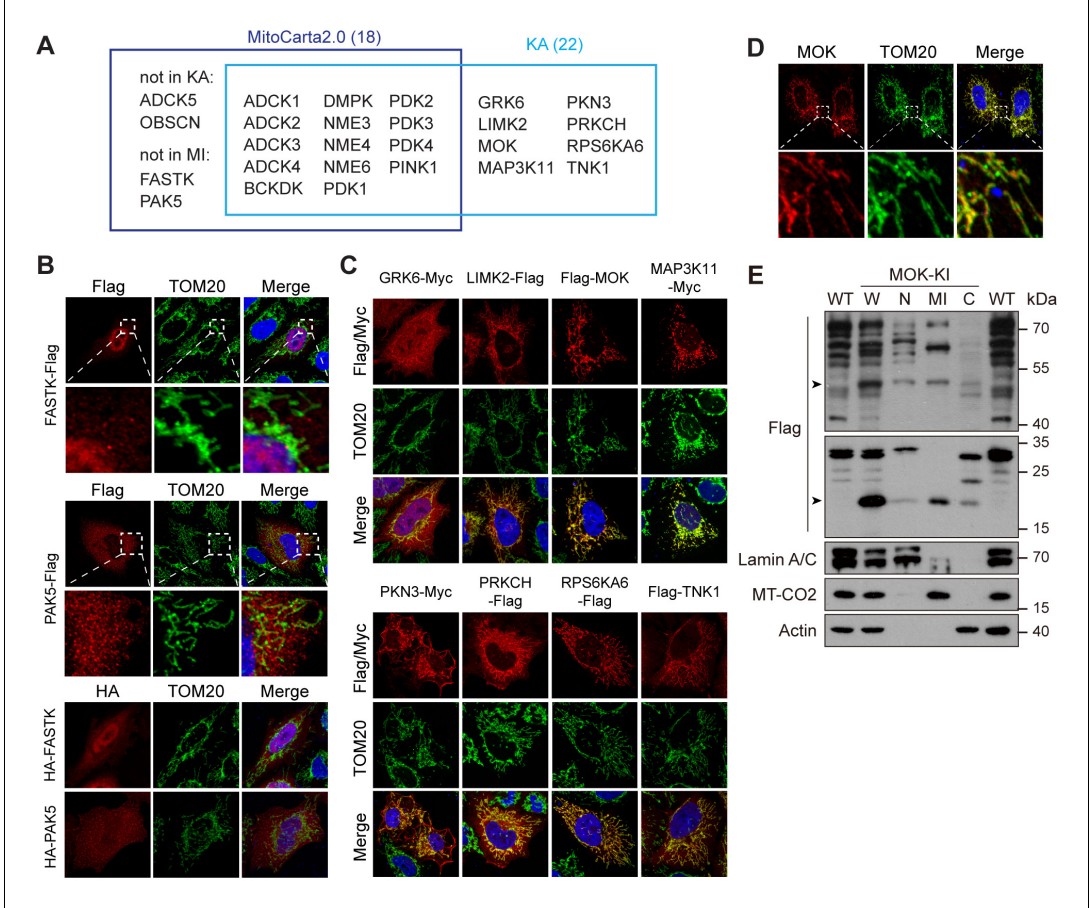

**Figure 5.** Kinome Atlas (KA) identified novel mitochondrial kinases. (**A**) Comparison of MitoCarta2.0 and KA. (**B**) FASTK and PAK5 are not mitochondrial. N- and C-terminal tagged kinases were expressed in HeLa cells and stained. (**C**) Mitochondrial kinases identified by KA. Experiments are similar to (**B**). (**D**) Overexpressed tagless MOK localized to mitochondria. Transfected HeLa cells were fixed and stained with anti-MOK antibody. (**E**) Endogenous MOK was present in mitochondria. MOK 3x Flag knock-in HeLa cells were fractionated and examined by western blotting. W: whole cell lysate; N: nucleus; MI: mitochondria; C: cytosol. Lamin A/C, MT-CO2, and Actin are nuclear, mitochondrial, and cytosolic markers, respectively. Equal loading was achieved by normalizing to the same number of cells. Arrowheads indicate MOK (*Figure 5—figure supplement 1*).

The online version of this article includes the following source data and figure supplement(s) for figure 5:

**Source data 1.** Uncropped western blot for *Figure 5*.

**Figure supplement 1.** Kinome Atlas (KA) identified novel mitochondrial kinases.

**Figure supplement 1—source data 1.** Uncropped western blot for *Figure 5—figure supplement 1*.

**Figure supplement 1—source data 2.** Raw data for *Figure 5—figure supplement 1*.

was further enhanced by moving epitope tags to the C-terminal (*Figure 5—figure supplement 1D*). We further searched HPA and found 11 mitochondrial kinases not recorded by MitoCarta2.0 (*Figure 5—figure supplement 1E*). Among them, AATK and SGK110 were not in the kinome library, while GRK6 and RPS6KA6 were confirmed by KA. Seven kinases were confirmed non-mitochondrial (*Figure 5—figure supplement 1F*). Although not in the above databases, MAP3K11 have been reported as a cell cycle-regulated kinase with mitochondria-like localization (*Swenson et al., 2003*). Taken together, KA revealed five new mitochondrial kinases, MOK, LIMK2, PKN3, PRKCH, and TNK1. We confirmed the mitochondrial localization of endogenous RPS6KA6 in HeLa cells using the HPA antibody (*Figure 5—figure supplement 1G*). However, tested antibodies for other kinases could not detect the endogenous proteins. Nevertheless, anti-MOK antibody confirmed ectopically expressed tagless MOK in mitochondria (*Figure 5D*). We further generated a MOK C-terminal 3x Flag knock-in HeLa strain (*Figure 5—figure supplement 1H*). siRNAs confirmed that a full-length 50 kDa band and a 20 kDa band were specific to MOK in western blotting (*Figure 5—figure supplement 1H, I*). By cellular fractionation, we confirmed that both isoforms were enriched in mitochondrial fractions with some presence in nucleus and cytosol (*Figure 5E*).

## MOK is a mitochondrial intermembrane space-localized kinase

Mitochondria have four sub-compartments with distinct functions, outer membrane (OMM), inner membrane (IMM), intermembrane space (IMS), and matrix (*Shimizu, 2019*). Using 3D structured illumination microscopy (3D-SIM), we found that signals of MOK, TNK1, and RPS6KA6 were wrapped around by TOM20, an OMM protein (*Figure 6A*), indicating transportation into mitochondria. In contrast, LIMK2 was in puncta partially colocalizing with TOM20, indicating that LIMK2 decorated the outer surface of mitochondria. LIMK2 has three splicing isoforms, 2a, 2b, and v1 (*Figure 6—figure supplement 1A*). LIMK2v1 in the kinome library lacks the classical function in inducing ectopic F-actin bundle (*Amano et al., 2001*; *Figure 6—figure supplement 1B*). Nevertheless, the F-actin-promoting LIMK2b also partially colocalized with mitochondria (*Figure 6—figure supplement 1B*). Therefore, LIMK2 may serve as a link between F-actin and mitochondrial function.

Several pieces of evidence identified MOK as an IMS protein. First, while mitochondrial MOK was visualized after cell permeabilization by Triton X-100, which breached both PM and OMM, it was not detected after cell permeabilization by digitonin, a detergent that penetrates PM but leaves OMM intact at limited concentration (*Vercesi et al., 1991*; *Figure 6B*). Thus, MOK was inside mitochondria. Second, extraction of isolated mitochondria using high pH hypotonic buffer released MOK as well as IMS protein EndoG, matrix protein HSP60, but not membrane components (*Figure 6C*). Thus, excluded MOK as a mitochondrial membrane protein. In addition, Hessian SIM super-resolution live-cell imaging (*Huang et al., 2018*) found that MOK-GFP wrapped around both IMM as marked by Mitotracker and matrix as marked by HSP60-mCherry (*Figure 6D*). The above data revealed MOK as an IMS protein. Noteworthy, the signal of MOK was discontinuous and enriched in puncta reminiscent of the Mitochondrial Contact Site and Cristae Organizing System (*Jakobs and Wurm, 2014*; *Figure 6D*), suggesting roles of MOK in mitochondrial structure.

We next determined the sequence basis for MOK mitochondrial localization. We confirmed that MOK isoform 1, longer than the isoform 2 as in the library (*Figure 6—figure supplement 1C*), was also present in mitochondria, although at a lower ratio (*Figure 6E*). Surprisingly, both an N-terminal fragment of MOK containing the kinase domain and a C-terminal fragment exhibited mitochondrial localizations similar to the wildtype protein (*Figure 6F*), suggesting at least two sequences mediating mitochondrial importing. Deletion of amino acids 53–103, but not other N-terminal fragments, largely eliminated the mitochondrial localization (*Figure 6F, Figure 6—figure supplement 1D*). In the C-terminal fragment, further truncating 30 or 60 amino acids (331–389, 361–389) eliminated mitochondria localization, and a peptide comprising only amino acids 301–360 had weak mitochondrial localization (*Figure 6F*). Therefore, two peptides 53–103 and 301–330 of MOK2 mediated the mitochondrial localization of MOK (*Figure 6—figure supplement 1C*). However, by sequence comparison to other new mitochondrial kinases, no new linear motif for mitochondrial localization was found.

Classically, mitochondrial importing of nuclear-encoded proteins relies on multi-protein translocator complexes such as the TOM complex, which imports precursors containing mitochondrial targeting sequences across OMM, and translocase of inner membrane (TIM) complex, which sorts precursors into matrix or IMM (*Schmidt et al., 2010*). shRNA-mediated knockdown of precursor

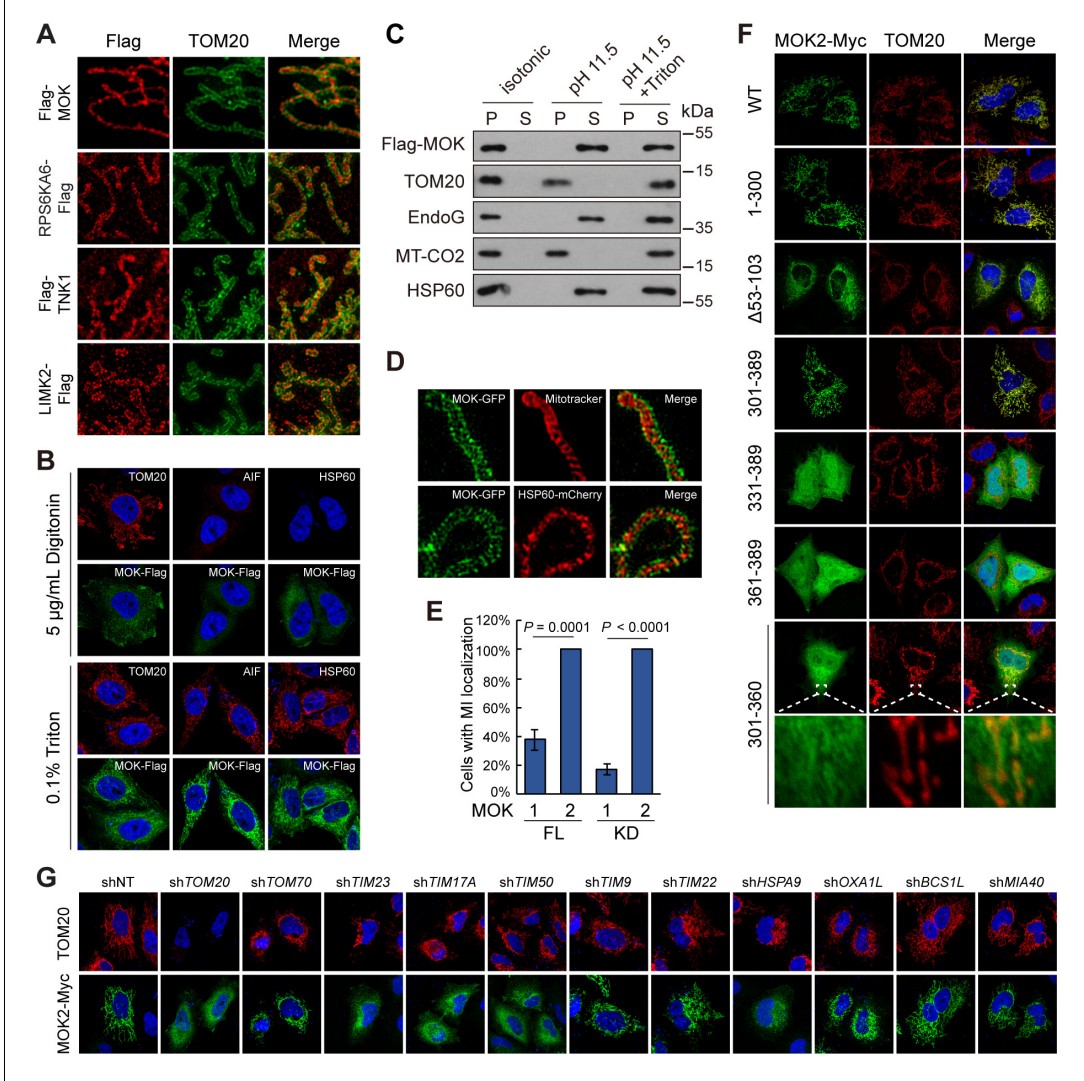

**Figure 6.** MOK is a mitochondrial intermembrane space (IMS)-localized protein. (**A**) Sub-mitochondrial localizations of new mitochondrial kinases. Transfected HeLa cells were stained and imaged by 3D structured illumination microscopy (3D-SIM). (**B**) MOK2 is imported into mitochondria. Transfected HeLa cells were fixed and permeabilized with digitonin or Triton X-100 before staining. (**C**) MOK is not a mitochondrial membrane protein. Mitochondria purified from transfected HeLa cells were incubated in indicated buffers, and then centrifuged. Supernatants (S) and pellets (P) were analyzed by western blotting. Equal loading was achieved by normalizing to the same number of cells. (**D**) MOK is an IMS protein. COS-7 cells transfected with MOK2-GFP were imaged by Hessian SIM. MitoTracker Red marked inner membrane (IMM) and co-transfected HSP60-mCherry marked matrix. (**E**) MOK2 has better mitochondrial localization than MOK1. HeLa cells were transfected with full-length (FL) or kinase domain (KD) of MOK1 and MOK2. Cells with mitochondrial MOK were quantified. Data is represented as mean ± SD; n = 3 independent experiments, two-tailed Student's *t* test; n = 100 cells were analyzed in each experiment. (**F**) Mitochondrial localization of MOK mutants. Transfected HeLa cells were stained with anti-Myc and anti-TOM20 antibodies. (**G**) TOM20-TIM23 complex is required for mitochondrial importing of MOK. HeLa cells expressing specific shRNAs were transfected with MOK2-Myc and stained (*Figure 6—figure supplement 1*).

The online version of this article includes the following source data and figure supplement(s) for figure 6:

**Source data 1.** Uncropped western blot for *Figure 6*.

**Source data 2.** Raw data for *Figure 6* and *Figure 6—figure supplement 1*.

**Figure supplement 1.** MOK is a mitochondrial intermembrane space (IMS)-localized protein.

**Figure supplement 1—source data 1.** Uncropped western blot for *Figure 6—figure supplement 1*.

receptor *TOM20* but not *TOM70* blocked mitochondrial localization of MOK, suggesting TOM20 as a receptor for MOK importing (*Figure 6G, Figure 6—figure supplement 1E*). Furthermore, knockdown of *TIM23, TIM17A,* and *TIM50* in the same complex, but not *TIM22* and *TIM9*, eliminated mitochondrial MOK (*Figure 6G*). Knockdown of *HSPA9*, which is responsible for translocation of matrix protein downstream of the TIM23 complex, also suppressed MOK mitochondrial localization (*Figure 6G*). These observations suggest a mechanism involving coupled TOM20 OMM receptor and TIM23 IMM channel under facilitation by HSPA9. In this case, MOK must be reversely transported to IMS or be cleaved by proteases before entry into the matrix. Since mitochondrial MOK had a similar molecular weight as cytoplasmic MOK (*Figure 5E*), a cleavage is unlikely. We therefore examined the role of OXA1L and BCS1L, which are responsible for export of matrix proteins in yeast (*Stiller et al., 2016*; *Wagener et al., 2011*). However, knockdown of *OXA1L* or *BCS1L* did not affect mitochondrial or sub-mitochondrial localizations of MOK (*Figure 6G, Figure 6—figure supplement 1F*). Furthermore, mitochondrial MOK was also insensitive to knockdown of *MIA40* (*Figure 6G*), which facilitates import of proteins into IMS through an oxidation-dependent mechanism (*Chacinska et al., 2004*). Nevertheless, by co-immunoprecipitation, we confirmed that MOK and its kinase domain physically interacted with TOM20 and TIM23 (*Figure 6—figure supplement 1G*). Taken together, MOK is an IMS protein imported by a TOM20-TIM23-dependent mechanism independent of classical mitochondrial importing signal.

## MOK regulates mitochondria structure and functions

To determine functions of MOK, we generated *MOK* knockout cells using CRISPR/Cas9 (*Figure 7—figure supplement 1A*). Electron microscopy revealed that while mitochondria length remained normal, the number of cristae markedly reduced in *MOK* KO cells (*Figure 7A, B*). This defect could be rescued by re-expression of MOK1 or MOK2. In Caki-1 cells, knockdown of *MOK* by shRNAs also resulted in loss of cristae (*Figure 7—figure supplement 1B–D*). Cristae have a pivotal role in generating ATP (*Cogliati et al., 2013*). Indeed, oxygen consumption rates (OCRs) for both basal respiration and ATP production were reduced in *MOK* KO cells (*Figure 7C, D*). Re-expression of MOK2 rescued mitochondrial respiration, while rescue by MOK1 was statistically insignificant. Consistently, knockout of *MOK* reduced cellular ATP level, which could be rescued by MOK2, but not MOK1 (*Figure 7E*). In Caki-1 cells, knockdown of MOK also reduced cellular ATP level (*Figure 7—figure supplement 1E*). Mitochondria are also a major origin of cellular reactive oxygen species (ROS). Under basal condition, ROS level was higher in *MOK* KO cells, but was rescued by expression of MOK1 and MOK2 (*Figure 7F*). Furthermore, oxidative stress induced by diamide caused more apoptosis in *MOK* KO cells (*Figure 7G*). Deregulated oxidative stress response was also observed in *MOK* KO MCF10A cells (*Figure 7—figure supplement 1F–H*).

We further determined the underlying mechanism of mitochondria regulation by MOK. Using in vitro autophosphorylation as an assay, we confirmed that MOK2 was kinase inactive (*Figure 7—figure supplement 1I*), consistent with its incomplete kinase domain. Nevertheless, MOK2 not only had better mitochondrial localization than MOK1, but also rescued mitochondrial structure and functions in KO cells. Thus, MOK played a functional role in mitochondria in an activity-independent manner. We next identified MOK-interacting proteins by tandem affinity purification (TAP) from MCF10A cells (*Figure 7H*). Mass spectrometry revealed that about 20% of proteins co-purified with MOK were mitochondrial proteins, including mitochondrial ribosomal proteins, solute carriers, oxidative phosphorylation-related proteins, and chaperones (*Figure 7—figure supplement 1J*). Immunoprecipitation confirmed interactions of MOK with IMM protein ATAD3A and solute carriers SLC25A13 and SLC25A11 (*Figure 7I, Figure 7—figure supplement 1K*). ATAD3A spans OMM and IMM with two transmembrane regions (*Gilquin et al., 2010*), and has important functions in mitochondrial structure dynamics (*Cooper et al., 2017*; *He et al., 2012*; *Zhao et al., 2019*). Indeed, ATAD3A knockdown cells also exhibited reduced cristae number (*Figure 7J, K, Figure 7—figure supplement 1L*). However, knockdown of ATAD3A did not worsen oxidative stress, suggesting that this function was mediated by other effectors (*Figure 7—figure supplement 1M*).

## Discussion

Here, we present Kinome Atlas (KA), a consolidated image database of the kinome on subcellular level with compartment annotations (http://cellimagelibrary.org/pages/kinome_atlas,

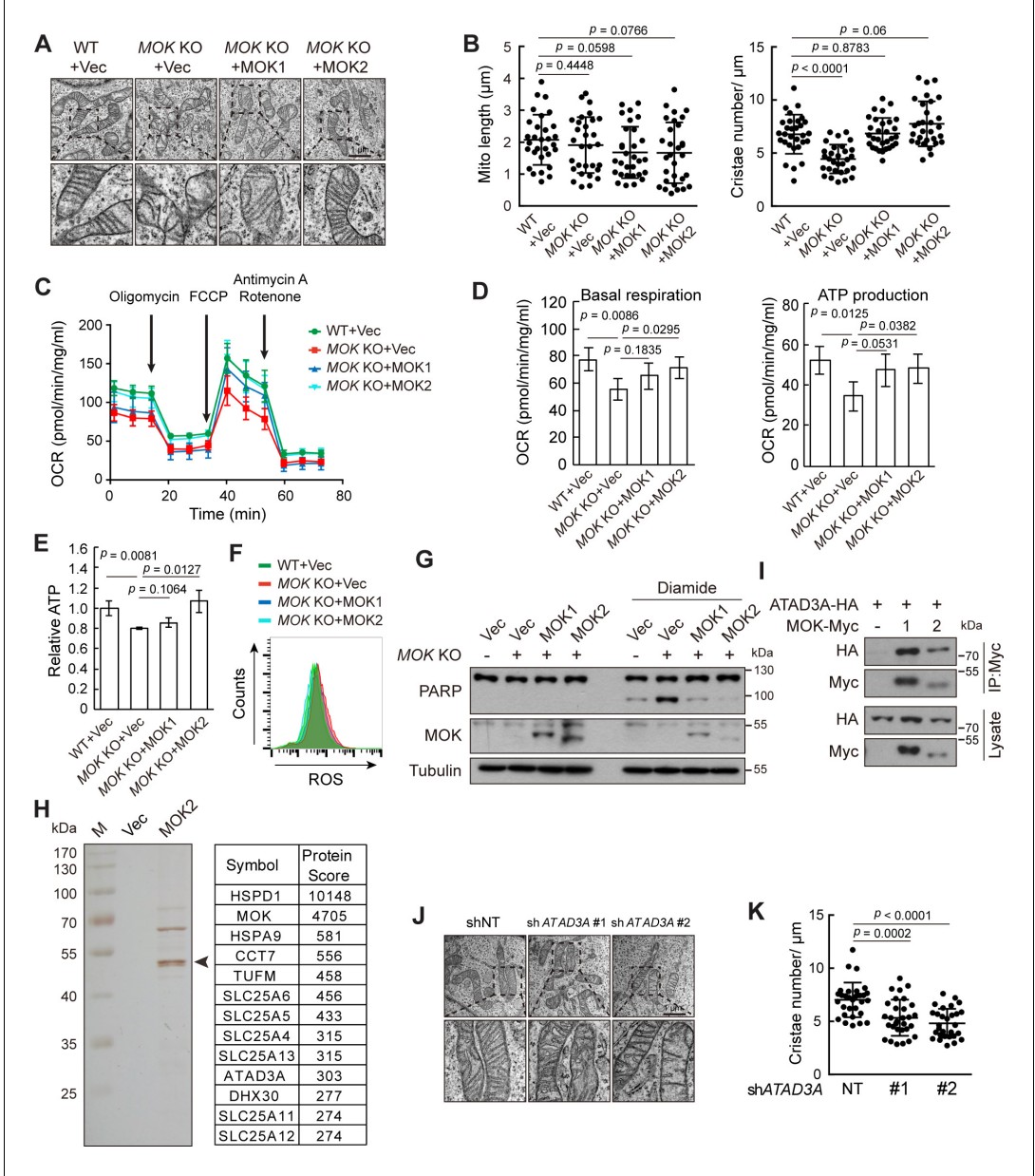

**Figure 7.** Knockout (KO) of *MOK* impairs mitochondrial structure and functions. (**A, B**) *MOK* KO reduced mitochondria cristae. Indicated A375 cells were examined by electron microscopy (**A**). The length and cristae number of mitochondria (n = 30) were quantified (**B**). (**C, D**) *MOK* KO impairs mitochondrial respiration. Cells sequentially exposed to mitochondrial inhibitors were analyzed on a Seahorse Analyzer. Oxygen consumption rate (OCR) was measured and normalized to protein concentration (**C**). Respiratory parameters were calculated from OCR data as described in the Materials and methods section (**D**). Data is representative and represented as mean ± SD, n = 3, two-tailed Student's *t* test. (**E**) *MOK* KO reduced cellular ATP level. Data is representative and represented as mean ± SD, n = 3, two-tailed Student's *t* test. (**F**) *MOK* KO increased cellular reactive oxygen species (ROS) level. Cells were stained with CM-H2DCFDA followed by flow cytometry. (**G**) *MOK* KO promoted apoptosis in response to oxidative stress. Indicated A375 cells were treated with 200 nM diamide for 20 hr, and cell lysates were analyzed by western blotting. (**H**) TAP identified MOK2 interacting proteins in mitochondria. MOK2-Flag-SBP was expressed in MCF10A cells, purified, and analyzed by silver staining after electrophoresis (left). Arrowhead denotes MOK. Top hits from mass spectrometry are shown (right). (**I**) MOK interacts with ATAD3A. HEK293T cells were transfected and cell lysates were immunoprecipitated with anti-Myc antibody. (**J, K**) Knockdown of *ATAD3A* reduced mitochondrial cristae number. Experiments were similar to these in (**A, B**). Data is representative and represented as mean ± SD, n = 30, two-tailed Student's *t* test (*Figure 7—figure supplement 1*).

The online version of this article includes the following source data and figure supplement(s) for figure 7:

**Source data 1.** Uncropped western blot for *Figure 7*.

**Source data 2.** Raw data for *Figure 7* and *Figure 7—figure supplement 1*.

*Figure 7 continued on next page*

*Figure 7 continued*

**Figure supplement 1.** Knockout of *MOK* impairs mitochondrial structure and functions.

**Figure supplement 1—source data 1.** Uncropped western blot for *Figure 7—figure supplement 1*.

**Figure supplement 1—source data 2.** Raw data for *Figure 7—figure supplement 1J*.

*Supplementary file 2*). Experiments in the same cell line transfected and cultured in a similar condition allowed comparison of individual kinases and kinase families. 19.1% (87/456) of kinases had only one localization, 72.4% (330/456) had a specific major localization (score ≥5), and 8.5% (39/456) were multi-localizing. While classical signal transduction pathways begin from signaling sensing at the plasma membrane, followed by signal transduction in the cytosol, and further to transcriptional regulation in the nucleus, we found enrichment of TK and AGC families on the plasma membrane, STE family in the cytosol, and CMGC, atypical, as well as CAMK families in the nucleus, suggesting division of roles between families. Consistently, small families are also enriched in organelles with more distinct functions, such as the enrichment of atypical kinase family in mitochondria, and over-representation of the RGC and PKL families in ER/Golgi. This distribution reflects the spatial dynamics of kinases as key molecules in signal transduction. It should be noted that due to the use of epitope tags and overexpression mislocalization is possible. However, KA will direct further validation of kinase subcellular localizations at the endogenous level in their physiologically relevant contexts. In relation to that, it should also be noticed that many kinases exhibit multiple subcellular localizations at different ratios, and other physiologically relevant localizations are also possible under a different context. It would be important to further investigate translocations of the kinome in response to various stimuli.

KA revealed kinases with previously unknown localizations to major or sub-compartments. Identification of SGK3 as an exosomal kinase suggests a role in intercellular communication. KA also found MAP4K3 in tubule structures overlapping with ER and microtubules, which is different from previous reports (*Hsu et al., 2018*). While the nature of this structure is not entirely clear, it is specifically regulated by LKB1, an upstream component of the mTOR pathway. It is possible that LKB1 regulates the structure itself or the localization of MAP4K3 to this structure. Intriguingly, MAP4K3 was reported to mediate the regulation of mTOR activity by amino acid sufficiency. Consistently, studies in *Drosophila* and mice indicated a role of MAP4K3 in regulation of body size (*Bryk et al., 2010*), immune response (*Chuang et al., 2011*), and longevity (*Chuang et al., 2019*), although the mechanism of action is not clear. Further elucidation of this LKB1-regulated localization of MAP4K3 may help solve the mystery. KA has also identified kinases enriched at nuclear peripheral regions, which may play a role in the maintenance and functions of this domain featured by gene-poor and compact chromatin with gene silencing markers (*Lieberman-Aiden et al., 2009*; *Stevens et al., 2017*). KA has also documented a collection of new kinases on plasma membrane, in ER, and in nucleolus, which all worth further studies. Except for cytosolic and nuclear localizations, prediction of these new localizations were largely ineffective, suggesting novel sorting mechanisms such as signal patches.

LLPS of proteins was recently recognized to play physiological roles in various contexts. Many protein kinases such as DYRK3, DYRK1A, PLK4, and FYN (*Amaya et al., 2018*; *Lu et al., 2018*; *Park et al., 2019*; *Wippich et al., 2013*), were reported to function in condensates formed by LLPS. In KA, we found that 38.2% of specific nuclear kinases had uneven localizations with puncta in varying size under overexpression condition. Furthermore, similar condensates were also found for cytoplasmic kinases. We did not rule out the possibility that overexpression was a reason for the frequent observation of kinase condensates in KA since LLPS is highly dependent on protein concentration (*Alberti et al., 2019*). Nevertheless, the fact that a high ratio of these kinases contains prion-like regions, and the deletion of which abolished condensate formation, suggests regulation with specificity. Importantly, at least for BMP2K, condensate formation was observed on the endogenous level. Kinase condensates formed by LLPS could regulate kinase activity. For instance, the yeast kinase Atg1 was found to form condensates through LLPS, which was important for autophagy initiation (*Fujioka et al., 2020*). Furthermore, LLPS could also organize multi-component condensates for phosphorylation of substrates, for example, in the case of DYRK1A, in regulation of RNA polymerase II (*Lu et al., 2018*). Interestingly, BMP2K was mutated in the IDR in developmental dysplasia of the

hip (DDH) and high myopia diseases (*Liu et al., 2009*; *Zhao et al., 2017*), suggesting functional roles of LLPS in BMP2K regulation and functions.

Mitochondrial kinases regulate mitochondria structure and functions. For example, PINK accumulates on dysfunctional mitochondria and recruits Parkin in a kinase-dependent manner to mediate autophagic removal of damaged organelles (*Chen and Dorn, 2013*). In this report, KA systematically identified new mitochondrial kinases, including MOK, LIMK2, PKN3, PRKCH, and TNK1, and confirmed less well-studied RPS6KA6, MAP3K11, and GRK6. All these new mitochondrial kinases lack classical mitochondrial importing signal peptide. However, at least for MOK, the importing still depends on the TOM-TIM complex. Although we did not exclude the possibility of an indirect mechanism, physical interaction between MOK and the TOM-TIM complex suggests unrecognized importing functions. Compared with other tissues, the mRNA level of MOK is much higher in spermatocytes and round spermatids. Thus, it is interesting to speculate MOK may function in spermatogenesis, during which mitochondria are dramatically remodeled for efficient energy production to support sperm movements (*Moraes and Meyers, 2018*; *Otani et al., 1988*). In addition, elevated expression of MOK in cancer, such as in cholangiocarcinoma, and skin cutaneous melanoma (http://firebrowse.org/), also suggests a functional role in tumorigenesis.

Taken together, the kinome of a single cell is partitioned with family-based features. Although limited to a snapshot of the live cell, KA compared 456 kinases under a similar experimental condition, thus resolved spatial distribution of one of the largest protein superfamilies. This resource, coupled with phospho-proteome analysis of cellular organelles, could broadly facilitate dissection of regulatory mechanisms for various cellular activities in normal and disease states.

## Materials and methods

### Construction of the kinome plasmid library and other plasmids

The majority of clones were made by gateway technology using human open reading frame (ORF) entry clones from two sources, Center for Cancer Systems Biology (CCSB) of Harvard University (hORFeome v3.1), or Thermo Fisher (Ultimate ORF Clone LITE Collection). Clones from CCSB were recombined into lentiviral vectors pXC-C-Flag or pLX304-C-Flag with C-terminal Flag tag, and clones from Thermo Fisher were recombined into a lentiviral vector pXN-N-HA/Flag with N-terminal HA and Flag tags due to the intact stop codon in entry clones. In addition, we got 79 clones by individual cloning or request from other investigators to supplement the collection. Kinase identities were confirmed by sequencing from the two ends. Kinase plasmids are available for request from the core facility of the Life Sciences Institute of Zhejiang University by emailing to lsicf@zju.edu.cn.

Kinase cDNAs were sub-cloned into pEGFP-C1 vector for live-cell imaging or into pcDNA-C-Myc and pcDNA-N-HA vectors to change epitope tag position. MOK was sub-cloned into pcDNA-C-Myc, pLVX-C-Flag-SBP, pcDNA-C-GFP, and pQCXIH vectors. HSP60 was sub-cloned from Ultimate ORF Clone LITE Collection into pcDNA-C-mCherry and pcDNA-C-GFP vectors. TOM20, TIM23, SLC25A11, SLC25A13, and ATAD3A were sub-cloned from Ultimate ORF Clone LITE Collection into pcDNA-C-GFP and pLVX-C-HA vectors. SEC61B was sub-cloned from Ultimate ORF Clone LITE Collection into pLVX-N-GFP vector. Point mutations were generated using QuikChange Site-Directed Mutagenesis Kit (Agilent Technologies). Kinase inactive mutants were generated by mutating the ATP binding Lys to Arg.

### Tissue culture, transfection, and viral infection

A375, COS-7, HEK293T, A549, DU 145, and HeLa cells were cultured in Dulbecco's Modified Eagle Medium (Thermo Fisher) containing 10% FBS (Thermo Fisher) and 50 µg/ml penicillin/streptomycin (P/S) (Thermo Fisher). U-2 OS, H460, and Caki-1 cells were cultured in RPMI-1640 (Thermo Fisher) containing 10% FBS and 50 µg/ml P/S. MCF10A cells were cultured in DMEM/F12 (Thermo Fisher) supplemented with 5% horse serum (Thermo Fisher), 20 ng/ml EGF, 0.5 µg/ml hydrocortisone, 10 µg/ml insulin, 100 ng/ml cholera toxin, and 50 µg/ml P/S. All cells were cultured in a 37°C humidified incubator with 5% $CO_2$. Caki-1 cells were from Dr. Liping Xie, and other cell lines were purchased from ATCC. Cell lines were authenticated using Short Tandem Repeat profiling at the Genetic Testing Biotechnology (Suzhou). Mycoplasma test for cell culture was done in a monthly basis using

MycoPlasma Detection Kit (Vazyme Biotech). Cells used in experiments were within 10 passages from thawing.

Transfection of plasmids was performed using Lipofectamine (Thermo Fisher), and transfection of siRNA was performed using Lipofectamine RNAiMAX (Thermo Fisher) according to the manufacturer's instructions. Lentiviral or retroviral infection was used to generate stable cells. Briefly, HEK293T cells were co-transfected with viral vectors and packaging plasmids. 48 hr post-transfection, virus-containing medium was collected, filtered through a 0.45 µm filter, and used to infect target cells in the presence of 5 µg/ml polybrene. Puromycin (Thermo Fisher) or Hygromycin B (Roche) at appropriate concentrations was used for selection.

## Immunofluorescent staining and imaging

Cells were seeded and transfected on glass coverslips. 24 hr post-transfection, cells were fixed with 4% paraformaldehyde in PBS for 15 min, permeabilized by 0.1% Triton X-100 (or 5 µg/ml digitonin as indicated) for 5 min, blocked in PBS containing 2% goat serum and 2% BSA for 30 min, and incubated with primary antibodies in blocking solution for 2 hr at room temperature. After washes, cells were stained with fluorophore-conjugated secondary antibodies (Thermo Fisher) for 1 hr at room temperature, and then further washed. Coverslips were mounted on glass slides in ProLong Gold antifade mounting media with DAPI (Thermo Fisher) and sealed. Pictures were taken by an LSM 710 confocal microscope (ZEISS). Super-resolution images were taken by 3D-SIM using Deltavision OMX (GE Healthcare) or Airyscan using LSM 880 confocal microscope (ZEISS). To determine sub-mitochondrial localization of MOK2, transfected COS-7 cells were stained with 500 nM MitoTracker Red CMXRos and monitored with Hessian SIM microscopy as previously described (*Huang et al., 2018*). Time-lapse live-cell imaging was performed with a DeltaVision Elite Cell Imaging System (GE Healthcare).

## Presentation on kinome tree

KA were presented on kinome tree using a web-based tool KinMap (*Eid et al., 2017*). Briefly, the localization and enrichment levels were reflected by colors and sizes of circles.

## CRISPR/Cas9-mediated gene knockout and knock-in

To generate *MOK* knockout cells, sgRNAs were cloned into PEP-KO vector. Transfected A375 and MCF10A cells were selected by puromycin for 48 hr. The remaining cells were suspended and re-seeded into single clones. After culture for 10 days, individual clones were picked into 24-well plate and amplified. *MOK* knockout clones were characterized by genomic DNA sequencing.

To perform epitope tag knock-in, a pair of sgRNAs targeting stop codon flanking regions were cloned into PEP-KI vector (derived from PEP-KO vector by replacing wild-type Cas9 with a D10A mutant). Then genomic regions flanking the stop codon were amplified by PCR to generate right and left homology arms. The homology arms were cloned into pSEPT vector using AgeI/NheI and EcoRI/SalI sites, respectively, and spaced by a 3x Flag coding sequence, a T2A element, and a neomycin resistance gene. One guanine in the protospacer adjacent motif (NGG) on one homology arm was mutated without altering amino acid coding to avoid cleavage. Cells transfected with all three plasmids were seeded as single clones and selected with G418 (Sangon Biotech). Knock-in clones were characterized by genotyping and western blotting.

## siRNA, shRNA, RNA extraction, and qPCR

For siRNA knockdown experiments, 30% confluent cells in 6-well plates were transfected with 20 nM siRNAs using Lipofectamine RNAiMAX (Thermo Fisher) following the manufacturer's protocol. siRNAs against target genes were synthesized by RiboBio.

pLKO.1 vector was used to construct shRNA-expressing lentiviral plasmids by integrating oligonucleotides between AgeI and EcoRI sites. Lentiviral infection was described above.

To determine knockdown efficiency, total RNA was extracted using RNAiso Plus (TaKaRa Bio). RNA was reverse-transcribed using random hexamers (TaKaRa Bio) and the resulting cDNA was analyzed by qPCR with gene-specific primers in the presence of SYBR Green (Thermo Fisher). Relative expression of mRNAs was normalized to the expression of *β-actin*.

## Cellular fractionation

HeLa cells were scrapped down in low salt buffer (10 mM HEPES pH 7.9, 10 mM KCl, 0.1 mM EDTA, 0.1 mM EGTA) with protease and phosphatase inhibitors, incubated on ice for 2 min, and pushed through a 25 G needle 10 times. Unbroken cells and nucleus were removed by centrifugation at 1000 g for 5 min at 4℃, and membrane fraction was pelleted by centrifuging at 100,000 g for 30 min with an Optima Max-XP ultracentrifuge (Beckman Coulter Life Sciences).

To isolate mitochondria, cells were homogenized in the starting buffer (225 mM mannitol, 75 mM sucrose, and 30 mM Tris-HCl pH7.4) and the homogenate was centrifuged at 600 g for 5 min at 4℃ to separate nuclear fraction. Supernatant was collected and centrifuged at 7000 g for 10 min at 4℃ to isolate mitochondrial and cytosolic fractions. Nuclear and mitochondrial fractions were washed once and re-pelleted.

## Western blotting and immunoprecipitation

Western blotting was performed following standard protocol. Briefly, protein samples were loaded, separated by SDS-PAGE, and transferred onto PVDF membranes. After incubation with primary antibodies overnight at 4℃, membranes were washed and incubated with HRP-conjugated secondary antibodies for 1.5 hr at room temperature. Proteins were detected using an ECL Detection Reagent (Thermo Fisher).

For immunoprecipitation, cells were lysed in ice-cold mild lysis buffer (100 mM NaCl, 10 mM EDTA, 1% NP40, 10 mM Tris pH 7.5, 50 mM NaF, 1 mM $Na_3VO_4$, 0.1 mM DTT) supplemented with EDTA-free Complete Protease Inhibitor. Cell lysate was centrifugated at 12,000 g for 15 min at 4℃ to collect supernatant fraction. After incubation with desired antibodies with rotation for 2 hr at 4℃, protein G-Sepharose (GE Healthcare) was added for an additional 1.5 hr. Then samples were centrifugated, washed with mild lysis buffer four times, and boiled with SDS loading buffer.

## Detection of SGK3 in culture medium

Transfected HEK293T cells were cultured in medium supplemented with exosome-deprived FBS prepared by centrifugation at 100,000 g for 2 hr. BFA was added to medium as indicated 24 hr before harvest. Culture medium was collected, volume recorded, and then centrifugated at 500 g for 10 min, 2000 g for 10 min, and 12,000 g for 15 min in sequence to remove cell debris. Resulted supernatant was supplemented with protease inhibitor cocktail. To immunoprecipitate SGK3, 1% NP-40 was added, and anti-Flag M2 resin (Sigma-Aldrich) was incubated with supernatant for 2 hr at 4℃. To separate extracellular vesicles, supernatant was further centrifugated at 100,000 g for 2 hr. Pellets were washed in PBS and dissolved in SDS loading buffer.

## In vitro kinase activity assay

Kinases were immunoprecipitated from transfected HEK293T cells. After washed in wash buffer (200 mM NaCl, 40 mM HEPES pH 7.5) and kinase assay buffer (50 mM potassium acetate, 30 mM HEPES, 5 mM $MgCl_2$), purified kinases were incubated in 45 μl kinase assay buffer in the presence of 200 μM ATP-γ-S and 20 μM ATP at 30℃ for 1 hr with vortex. 2 μl EDTA (0.5 M) was used to stop reaction followed by additional 2 μl PNBM (50 mM) to start alkylation for 30 min at room temperature. Assay was completed by boiling reaction mixture at 95℃ for 5 min with SDS loading buffer.

## Alkaline carbonate extraction of mitochondria

Freshly isolated mitochondria were resuspended in isotonic buffer (250 mM sucrose, 10 mM KCl, 2 mM $MgCl_2$, 1 mM EDTA, 10 mM Tirs-HCl pH 7.4). Three groups were separately added with equal volume of isotonic buffer, 0.1 M sodium carbonate (pH 11.5), or 0.1 M sodium carbonate with 0.5% TritonX-100, respectively. After an incubation on ice for 30 min, samples were centrifuged at 13,000 g for 20 min at 4℃. Pellets were resuspended in isotonic buffer. All samples were analyzed by western blotting.

## Seahorse analysis

OCR was monitored using the Seahorse Analyzer XF96 (Agilent Technologies) according to the manufacturer's instructions. Briefly, 10,000 cells were seeded in each well and cultured in the XF96 culture plate while the XF cartridge was incubated overnight at 37℃ without $CO_2$. To begin analysis,

cells were washed twice and incubated in XF assay medium at 37℃ for 1 hr. 5 µM oligomycin A, 10 µM FCCP, and 5 µM rotenone/antimycin were used in OCR detection. After analysis, protein concentration was determined for normalization using a BCA Assay Kit (Thermo Fisher). Calculation of normalized OCR was performed following the manufacturer's instructions (basal respiration = last rate measurement before the first injection – minimum rate measurement after rotenone/antimycin A injection, ATP production = last rate measurement before oligomycin injection – minimum rate measurement after oligomycin injection).

## Measurement of cellular ATP level

ATP was measured using an ATP Assay Kit (Beyotime Biotechnology) according to the manufacturer's instructions. Briefly, cells were homogenized in ice-cold ATP-releasing buffer. After centrifugation, supernatants were added to substrate solution. A standard curve of ATP concentration (50 nM to 10 µM) was prepared from measurement of standard solutions. Luminescence was recorded by a microplate reader (POLARstar Omega). Protein concentration was determined for normalization using a BCA Assay Kit.

## Transmission electron microscopy

Cells growing on coverslips were washed in PBS, fixed with 2.5% glutaraldehyde for 2 hr at room temperature, followed by an overnight incubation in 2.5% glutaraldehyde at 4℃. After washes and fixation in 1% osmic acid for 1 hr, samples were washed in PBS, dehydrated in a series of gradient ethanol (50, 70, 80, 90, 95, and 100%), embedded in Epon Resin, and then cut into 60 nm ultrathin sections. After uranyl acetate and lead citrate staining, samples were observed with a Hitachi HT7700 electron microscope. Representative views were pictured and analyzed by an Image-Pro Plus 6.0 software.

## Measurement of cellular ROS level

Cells were trypsinized, washed, and re-suspended in PBS. After incubation with 5 µM CM-H2DCFDA for 10 min, cellular ROS level was detected by a flow cytometer (Beckman Coulter).

## Tandem affinity purification

MCF10A cells stably expressing MOK2-Flag-SBP were trypsinized and lysed in lysis buffer (10 mM Tris pH 7.5, 2 mM EDTA, 150 mM NaCl, 0.3% CHAPS, 2.5 mM NaF, 1 mM $Na_3VO_4$, 0.1 mM DTT, 0.5 mM PMSF) supplemented with EDTA-free Complete Protease Inhibitor. Lysate was cleared by centrifugation at 12,000 g for 15 min at 4℃, and then incubated with anti-Flag M2 resin (Sigma-Aldrich) for 2 hr at 4℃ and washed three times with lysis buffer. Bound proteins were eluted with lysis buffer containing 200 ng/µl 3x Flag peptide (Sigma-Aldrich). Eluate was further incubated with streptavidin-conjugated resin (Agilent Technologies) for 2 hr at 4℃. Resins were washed and then eluted with elution buffer (Tris, pH 7.5, 150 mM NaCl, 0.05% RapiGest, 10 mM 2-mercaptoethanol) containing 4 mM biotin. Samples were then processed for MS/MS analysis.

## Protein sequence analysis for IDR

Domain architecture of each kinase was analyzed by SMART (http://smart.embl.de/). Intrinsic disorder tendency was predicted by IUPred2A (https://iupred2a.elte.hu/) with scores assigned between 0 and 1. A score above 0.5 indicates disorder. Predicted IDRs and their boundaries were marked on domain architecture. Prion-like domains (PrD.like, red) of each kinase were predicted by PLAAC (http://plaac.wi.mit.edu/) with scores assigned between 0 and 1. A score above 0.5 indicates prion-like region. FoldIndex (gray), -PLACC (red), and −4*PAPA (green) were also produced by PLAAC (http://plaac.wi.mit.edu/). FoldIndex (gray) predicts whether a given protein sequence is intrinsically unfolded with scores assigned between −1 and 1. A score above 0 indicates folded region, while a score below 0 indicates unfolded region. Sliding averages of per-residue log-likelihood ratios for the prion-like versus background state were scaled by using base −4 logarithms and reversed in sign, thereby producing -PLACC (red). A prion aggregation prediction algorithm (PAPA) focusing on amino acid composition of proteins predicts prion propensity of Q/N-rich proteins. In order to be more comparable to the other tracks, PAPA multiplied by −4 produces 4*PAPA (green), so that

lower scores are more predictive of prion propensity. −4*PAPA below the dashed green line (−0.2) indicates prion propensity.

### Alignment and prediction of kinase localization in databases

Kinases with specific localization in KA were analyzed by comparing localization information with that from COMPARTMENTS and HPA. To perform localization alignment with COMAPRTMENTS, localization information with confidence scores more than three stars (localization information adopted from HPA was omitted) from knowledge channel was extracted and aligned with one localization in KA. The localization-matched kinases were those sharing the specific localization with KA. Localization alignment with HPA was performed in a similar manner except main localization in HPA was used.

The predictions channel in COMPARTMENTS contains predicted results from WoLF PSORT and YLoc. To perform kinase localization prediction, top three predicted localizations were extracted and aligned with KA as described above.

### Quantification and statistical analysis

No statistical methods were used to estimate sample size. A standard two-tailed unpaired Student's *t* test was used for statistical analysis of two groups. p values lower than 0.05 were considered statistically significant. We performed statistical analyses using GraphPad Prism. Heatmaps were drawn using R package. Venn diagrams were drawn using web-based VENNY2.1.

## Acknowledgements

We thank Drs. Xing Guo, Jun Huang, Hai Song, and Junyu Xiao for plasmids; Dr. Yongchao Zhao for support with Seahorse analysis; Dr. Jinrong Peng for the UTP25 antibody; and Dr. Liping Xie for Caki-1 cells. We thank the core facility of the Life Sciences Institute for technical assistance, especially Weina Shang for support with imaging. This work was supported by grants to B Zhao from the National Natural Science Foundation of China General Project (31970726), Key Project (81730069), the National Key R&D Program of China (2017YFA0504502), Natural Science Foundation of Zhejiang Key Project (LZ21C070002), Key Laboratory of Growth Regulation and Translational Research of Zhejiang Province Open Project, and the Fundamental Research Funds for the Central Universities.

## Additional information

### Funding

| Funder | Grant reference number | Author |
| --- | --- | --- |
| National Natural Science Foundation of China | 31970726 | Bin Zhao |
| National Natural Science Foundation of China | 81730069 | Bin Zhao |
| Ministry of Science and Technology of the People's Republic of China | 2017YFA0504502 | Bin Zhao |
| Natural Science Foundation of Zhejiang Province | LZ21C070002 | Bin Zhao |
| Key Laboratory of Growth Regulation and Translational Research of Zhejiang Province | 2020E10027 | Bin Zhao |
| Fundamental Research Funds for the Central Universities | | Bin Zhao |

The funders had no role in study design, data collection and interpretation, or the decision to submit the work for publication.

## Author contributions

Haitao Zhang, Formal analysis, Validation, Investigation, Writing - original draft; Xiaolei Cao, Mei Tang, Formal analysis, Validation, Investigation; Guoxuan Zhong, Yuan Si, Haidong Li, Feifeng Zhu, Qinghua Liao, Jianhui Zhao, Jia Feng, Shuaifeng Li, Chenliang Wang, Validation, Investigation; Liuju Li, Resources, Investigation, Methodology; Manuel Kaulich, Resources; Fangwei Wang, Li Li, Zongping Xia, Tingbo Liang, Xin-Hua Feng, Resources, conceptual advice; Liangyi Chen, Resources, Methodology; Huasong Lu, Resources, Investigation, conceptual advice; Bin Zhao, Conceptualization, Funding acquisition, Writing - original draft, Writing - review and editing

## Author ORCIDs

Haidong Li http://orcid.org/0000-0002-4427-7920
Jia Feng http://orcid.org/0000-0002-6742-484X
Manuel Kaulich http://orcid.org/0000-0002-9528-8822
Fangwei Wang http://orcid.org/0000-0001-5617-282X
Liangyi Chen http://orcid.org/0000-0003-1270-7321
Bin Zhao https://orcid.org/0000-0002-1690-646X

## Decision letter and Author response

Decision letter https://doi.org/10.7554/eLife.64943.sa1
Author response https://doi.org/10.7554/eLife.64943.sa2

# Additional files

## Supplementary files

• Supplementary file 1. Human kinome and the kinome plasmid library, related to *Figure 1A, B*, *Figure 1—figure supplement 1A, B*, and *Figure 2A*.

• Supplementary file 2. Summary of the Kinome Atlas, related to *Figure 1E–G*, *Figure 1—figure supplement 1D*, and *Figure 2—figure supplement 1C*.

• Supplementary file 3. Coverage of the kinome by Kinome Atlas (KA) and databases, related to *Figure 2A*.

• Supplementary file 4. Kinome Atlas (KA) in comparison with COMPARTMENTS, related to *Figure 2C, D*.

• Supplementary file 5. Kinome Atlas (KA) in comparison with Human Protein Atlas (HPA), related to *Figure 2E, F*, *Figure 2—figure supplement 1D, E*.

• Supplementary file 6. Analysis of kinases uniquely annotated by Kinome Atlas (KA), related to *Figure 2G,H* and *Figure 3*.

• Transparent reporting form

## Data availability

Please use this version in the DAS:All data generated or analyzed during this study are included in the manuscript and supporting files. Images of KA are available at the Cell Image Library database http://cellimagelibrary.org/pages/kinome_atlas.

The following dataset was generated:

| Author(s) | Year | Dataset title | Dataset URL | Database and Identifier |
|---|---|---|---|---|
| Zhang H, Cao X, Tang M, Zhong G, Si Y, Li H, Zhu F, Liao Q, Li L, Zhao J, Feng J, Li S, Wang C, Kaulich M, Wang F, Chen L, Xia Z, Liang T, Lu H, Feng XH, Zhao B | 2021 | The Kinome Atlas | http://cellimagelibrary. org/pages/kinome_atlas | Cell Image Library, kinome_atlas |

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

# Appendix 1

## Key resources table related to the 'Materials and methods' section

**Appendix 1—key resources table**

| Reagent type (species) or resource | Designation | Source or reference | Identifiers | Additional information |
|---|---|---|---|---|
| Gene (*Homo sapiens*) | Human kinome cDNA | Center for Cancer Systems Biology of Harvard University | hORFeome v3.1 | |
| Gene (*Homo sapiens*) | Human kinome cDNA | Thermo Fisher | Ultimate ORF Clone LITE Collection | |
| Cell line (*Homo sapiens*) | HeLa | ATCC | Cat# CCL-2, RRID:CVCL_0030 | |
| Cell line (*Homo sapiens*) | HEK293T | ATCC | Cat# CRL-11268, RRID:CVCL_1926 | |
| Cell line (*Homo sapiens*) | A375 | ATCC | Cat# CRL-1619, RRID:CVCL_0132 | |
| Cell line (*Homo sapiens*) | DU 145 | ATCC | Cat# HTB-81, RRID:CVCL_0105 | |
| Cell line (*Homo sapiens*) | MCF10A | ATCC | Cat# CRL-10317, RRID:CVCL_0598 | |
| Cell line (*Homo sapiens*) | Caki-1 | Liping Xie (Zhejiang University) | | |
| Cell line (*Cercopithecus aethiops*) | COS-7 | ATCC | Cat# CRL-1651, RRID:CVCL_0224 | |
| Transfected construct (*Homo sapiens*) | siNT | This paper | | Control siRNA target sequence: UUCUCCGAACGUGUCACGU |
| Transfected construct (*Homo sapiens*) | si*MOK#1* | This paper | | siRNA target sequence: CCUCUUUCCUGGAGUAAAU |
| Transfected construct (*Homo sapiens*) | si*MOK#2* | This paper | | siRNA target sequence: AGUCGAGAGCUAUGAAUUU |
| Transfected construct (*Homo sapiens*) | shNT | This paper | | Control shRNA target sequence: CCTAAGGTTAAGTCGCCCTCG |
| Transfected construct (*Homo sapiens*) | sh-*TOM20* | This paper | | shRNA target sequence: GGCGTAGACCATCTGACAAAT |
| Transfected construct (*Homo sapiens*) | sh-*TOM70* | This paper | | shRNA target sequence: GCATGCTGTTAGCCGATAAAG |

*Continued on next page*

*Appendix 1—key resources table continued*

| Reagent type (species) or resource | Designation | Source or reference | Identifiers | Additional information |
|---|---|---|---|---|
| Transfected construct (*Homo sapiens*) | sh-*TIM17A* | This paper | | shRNA target sequence: GCTGGTATCTTGTTGACAAGA |
| Transfected construct (*Homo sapiens*) | sh-*TIM50* | This paper | | shRNA target sequence: CCTCAAGACCATTGCACTGAA |
| Transfected construct (*Homo sapiens*) | sh-*TIM23* | This paper | | shRNA target sequence: CCAGCCTCTATGCACTATATA |
| Transfected construct (*Homo sapiens*) | sh-*HSPA9* | This paper | | shRNA target sequence: CGTGCTCAATTTGAAGGGATT |
| Transfected construct (*Homo sapiens*) | sh-*HSP60* | This paper | | shRNA target sequence: CCTGCTCTTGAAATTGCCAAT |
| Transfected construct (*Homo sapiens*) | sh-*MIA40* | This paper | | shRNA target sequence: TCTTGACATCTTGACATATAC |
| Transfected construct (*Homo sapiens*) | sh-*TIM9* | This paper | | shRNA target sequence: GCTTGGTCACTTGATTAGAAA |
| Transfected construct (*Homo sapiens*) | sh-*TIM22* | This paper | | shRNA target sequence: GCTTTGACCCTAAGGATCCTT |
| Transfected construct (*Homo sapiens*) | sh-*OXA1L* | This paper | | shRNA target sequence: CGAATCAGAGAGGCCAAGTTA |
| Transfected construct (*Homo sapiens*) | sh-*BCS1L* | This paper | | shRNA target sequence: CGTCCAGGAATTCATCGATAA |
| Transfected construct (*Homo sapiens*) | sh-*MOK* #1 | This paper | | shRNA target sequence: CTGGTTCTCTTGCAC TAATAT |
| Transfected construct (*Homo sapiens*) | sh-*MOK* #2 | This paper | | shRNA target sequence: ACCTCTACTAACAACCAATTT |
| Transfected construct (*Homo sapiens*) | sh-*ATAD3A* #1 | This paper | | shRNA target sequence: CATCAATGAGATGGTCCACTT |
| Transfected construct (*Homo sapiens*) | sh-*ATAD3A* #2 | This paper | | shRNA target sequence: CAAGGACAAATGGAGCAACTT |
| Transfected construct (*Homo sapiens*) | sg-MOK KO #1 | This paper | | PEP-KO with gRNA sequence: GTACCAGTTATGTAAGTCCC |
| Transfected construct (*Homo sapiens*) | sg-MOK KO #2 | This paper | | PEP-KO with gRNA sequence: GCAATTGGCAAAATAGGAGA |
| Transfected construct (*Homo sapiens*) | sg-BMP2K KI #1 | This paper | | PEP-KI with gRNA sequence: GAAATGGAGCAGCACCAAAT |
| Transfected construct (*Homo sapiens*) | sg-BMP2K KI #2 | This paper | | PEP-KI with gRNA sequence: TAAACAGTAGATACTTCTGA |

*Continued on next page*

*Appendix 1—key resources table continued*

| Reagent type (species) or resource | Designation | Source or reference | Identifiers | Additional information |
|---|---|---|---|---|
| Transfected construct (*Homo sapiens*) | sg-MOK KI #1 | This paper | | PEP-KI with gRNA sequence: TCTTCCGCCTTTCCGCACTA |
| Transfected construct (*Homo sapiens*) | sg-MOK KI #2 | This paper | | PEP-KI with gRNA sequence: CACCGTCGTCTCGACTTCGG |
| Antibody | Mouse monoclonal anti-alpha 1 Sodium Potassium ATPase antibody | Abcam | Cat# ab7671, RRID:AB_306023 | WB (1:2000) |
| Antibody | Mouse monoclonal anti-GM130 antibody | BD Biosciences | Cat# 610823, RRID:AB_398142 | IF (1:1000) |
| Antibody | Rabbit polyclonal anti-UTP25 antibody | Dr. Jinrong Peng (Zhejiang University) | | IF (1:200) |
| Antibody | Rabbit polyclonal anti-Lamin A/C (H-110) antibody | Santa Cruz Biotechnology | Cat# sc-20681, RRID: AB_648154 | WB (1:2000) |
| Antibody | Mouse monoclonal anti-Lamin A/C (636) antibody | Santa Cruz Biotechnology | Cat# sc-7292, RRID:AB_627875 | IF (1:1000) |
| Antibody | Rabbit monoclonal anti-Catalase (D4P7B) antibody | Cell Signaling Technology | Cat# 12980, RRID:AB_2798079 | IF (1:100) |
| Antibody | Rabbit polyclonal anti-Pericentrin antibody | Covance | Cat# PRB-432C-200, RRID:AB_291635 | IF (1:500) |
| Antibody | Rabbit monoclonal anti-EEA1 (C45B10) antibody | Cell Signaling Technology | Cat# 3288, RRID:AB_2096811 | IF (1:100) |
| Antibody | Rabbit monoclonal anti-TOM20 (F-10) antibody | Santa Cruz Biotechnology | Cat# sc-17764, RRID: AB_628381 | IF (1:1000) |
| Antibody | Mouse polyclonal anti-TOM20 (FL-145) antibody | Santa Cruz Biotechnology | Cat# sc-11415, RRID: AB_2207533 | IF (1:1000) |
| Antibody | Rabbit monoclonal anti-HSP60 (D6F1) antibody | Cell Signaling Technology | Cat# 46611, RRID:AB_2799305 | WB (1:1000), IF (1:500) |
| Antibody | Mouse monoclonal anti-ENDOG (B-2) antibody | Santa Cruz Biotechnology | Cat# sc-365359, RRID: AB_10843802 | WB (1:2000) |
| Antibody | Mouse monoclonal anti-MT-CO2 antibody | Abcam | Cat# ab110258, RRID:AB_10887758 | WB (1:2000) |
| Antibody | Mouse monoclonal anti-LAMP1 antibody | Santa Cruz Biotechnology | Cat# sc-20011, RRID: AB_626853 | IF (1:500) |
| Antibody | Mouse monoclonal anti-HSP90 antibody | BD Biosciences | Cat# 610418, RRID:AB_397798 | WB (1:5000) |

*Continued on next page*

*Appendix 1—key resources table continued*

| Reagent type (species) or resource | Designation | Source or reference | Identifiers | Additional information |
|---|---|---|---|---|
| Antibody | Rabbit monoclonal anti-pErk1/2 (Thr202/Tyr204) antibody | Cell Signaling Technology | Cat# 4370, RRID:AB_2315112 | WB (1:2000) |
| Antibody | Rabbit monoclonal anti-Thiophosphate ester antibody | Abcam | Cat# ab133473, RRID:AB_2737094 | WB (1:5000) |
| Antibody | Rabbit monoclonal anti-PARP (46D11) antibody | Cell Signaling Technology | Cat# 9532, RRID:AB_659884 | WB (1:2000) |
| Antibody | Rabbit polyclonal anti-RPS6KA6 antibody | Sigma-Aldrich | Cat# HPA002852, RRID:AB_1079854 | IF (1:100) |
| Antibody | Rabbit polyclonal anti-MOK antibody | Sigma-Aldrich | Cat# HPA027292, RRID:AB_10600989 | WB (1:1000) |
| Antibody | Mouse monoclonal ANTI-FLAG M2 antibody | Sigma-Aldrich | Cat# F3165, RRID:AB_259529 | IF (1:1000) |
| Antibody | Rabbit monoclonal anti-DYKDDDDK Tag (D6W5B) antibody | Cell Signaling Technology | Cat# 14793, RRID:AB_2572291 | IF (1:500) |
| Antibody | Mouse monoclonal anti-Flag M2-Peroxidase-HRP conjugated antibody | Sigma-Aldrich | Cat# A8592, RRID:AB_439702 | WB (1:5000) |
| Antibody | Rabbit monoclonal anti-HA-Tag (C29F4) antibody | Cell Signaling Technology | Cat# 3724, RRID:AB_1549585 | IF (1:500) |
| Antibody | Mouse monoclonal anti-Myc-Tag (9B11) antibody | Cell Signaling Technology | Cat# 2276, RRID:AB_331783 | IF (1:500) |
| Antibody | Rabbit polyclonal anti-GFP antibody | Abcam | Cat# ab6556, RRID:AB_305564 | WB (1:2000) |
| Antibody | Mouse monoclonal anti-β-actin antibody | Proteintech | Cat# 66009–1-Ig, RRID:AB_2687938 | WB (1:5000) |
| Antibody | Mouse monoclonal anti-Tubulin (clone B5-1-2) antibody | Sigma-Aldrich | Cat# T6074, RRID:AB_477582 | WB (1:5000), IF (1:1000) |
| Antibody | Mouse monoclonal anti- γ-Tubulin (clone GTU88) antibody | Sigma-Aldrich | Cat# T5326, RRID:AB_532292 | IF (1:500) |
| Antibody | Goat polyclonal anti-Rabbit IgG(H + L)-Alexa Fluor 488 conjugated antibody | Thermo Fisher Scientific | Cat# A-11034, RRID:AB_2576217 | IF (1:1000) |
| Antibody | Goat polyclonal anti-Mouse IgG(H + L)-Alexa Fluor 488 conjugated antibody | Thermo Fisher Scientific | Cat# A-11029, RRID:AB_2534088 | IF (1:1000) |
| Antibody | Goat polyclonal anti-Rabbit IgG(H + L)-Alexa Fluor 594 conjugated antibody | Thermo Fisher Scientific | Cat# A-11037, RRID:AB_2534095 | IF (1:1000) |

*Appendix 1—key resources table continued*

| Reagent type (species) or resource | Designation | Source or reference | Identifiers | Additional information |
|---|---|---|---|---|
| Antibody | Goat polyclonal anti-Mouse IgG(H + L)-Alexa Fluor 594 conjugated antibody | Thermo Fisher Scientific | Cat# A-11032, RRID:AB_2534091 | IF (1:1000) |
| Antibody | Goat polyclonal anti-Rabbit IgG(H + L)-Alexa Fluor 647 conjugated antibody | Thermo Fisher Scientific | Cat# A-21244, RRID:AB_2535812 | IF (1:1000) |
| Sequence-based reagent | *TOM20* RT-F | This paper | | AGGGCGTAGACCATCTGACA |
| Sequence-based reagent | *TOM20* RT-R | This paper | | TTCAGCCAAGCTCTGAGCAC |
| Sequence-based reagent | *TOM70* RT-F | This paper | | AGACGTGCAAAAGCCCATGA |
| Sequence-based reagent | *TOM70* RT-R | This paper | | AAGCATGGGCTGGGAAATGA |
| Sequence-based reagent | *TIM17A* RT-F | This paper | | GGGAGGGCTGTTTTCCATGA |
| Sequence-based reagent | *TIM17A* RT-R | This paper | | GGAGGGGTCTTCTGCAAACT |
| Sequence-based reagent | *TIM50* RT-F | This paper | | TGCACGAGGTTGGCGA |
| Sequence-based reagent | *TIM50* RT-R | This paper | | TGTATGTCCGGCGCAACTG |
| Sequence-based reagent | *TIM23* RT-F | This paper | | AGGGGCACTTTGGGCTAATAC |
| Sequence-based reagent | *TIM23* RT-R | This paper | | TATCCCTCGAAGACCACCTGT |
| Sequence-based reagent | *HSPA9* RT-F | This paper | | CCTACGGCCTGGACAAGAAG |
| Sequence-based reagent | *HSPA9* RT-R | This paper | | CTTGTGCTTGCGCTTGAACT |
| Sequence-based reagent | *HSP60* RT-F | This paper | | CTTTTAGCCGATGCTGTGGC |
| Sequence-based reagent | *HSP60* RT-R | This paper | | TTGGCTATAGAGCGTGCCAG |
| Sequence-based reagent | *MIA40* RT-F | This paper | | CTATTGCCGGCAGGAAGGG |
| Sequence-based reagent | *MIA40* RT-R | This paper | | GGCATGGGCAGTTCCAGTTA |
| Sequence-based reagent | *TIM9* RT-F | This paper | | ACAGAGACCTGCTTTTTGGACT |
| Sequence-based reagent | *TIM9* RT-R | This paper | | GCCAGGGCTTCATTCTGCTG |
| Sequence-based reagent | *TIM21* RT-F | This paper | | CCTGAGACAGCGGGTTCC |
| Sequence-based reagent | *TIM21* RT-R | This paper | | AAGACAAATCCTCCCACGCA |
| Sequence-based reagent | *OXA1L* RT-F | This paper | | CTCGCAATGGCTTGGGAAAC |
| Sequence-based reagent | *OXA1L* RT-R | This paper | | CTCAGGTACTGCTGTGGGTG |

*Continued on next page*

*Appendix 1—key resources table continued*

| Reagent type (species) or resource | Designation | Source or reference | Identifiers | Additional information |
|---|---|---|---|---|
| Sequence-based reagent | *BCS1L* RT-F | This paper | | ATGGCGTCCCTTTGGCTATC |
| Sequence-based reagent | *BCS1L* RT-R | This paper | | AGTCGGTCATCAGAGAGGCT |
| Sequence-based reagent | *MOK* RT-F | This paper | | TGAGCTAATACGAGGGAGAAGA |
| Sequence-based reagent | *MOK* RT-R | This paper | | TACTCCAGGAAAGAGGGGCT |
| Sequence-based reagent | *ATAD3A* RT-F | This paper | | GCCTCCTGCTCTTTGTGGAT |
| Sequence-based reagent | *ATAD3A* RT-R | This paper | | AACTGCTCTGGTTGGTTGCT |
| Sequence-based reagent | *ACTB* RT-F | This paper | | CTCGCCTTTGCCGATCC |
| Sequence-based reagent | *ACTB* RT-R | This paper | | GAATCCTTCTGACCCATGCC |
| Commercial assay or kit | Seahorse XF Cell Mito Stress Test Kit | Agilent Technologies | Cat# 103015-100 | |
| Commercial assay or kit | ATP detection kit | Beyotime Biotechnology | Cat# S0026 | |
| Chemical compound, drug | ProLong Gold Antifade Mountant with DAPI | Thermo Fisher Scientific | Cat# P36931 | |
| Chemical compound, drug | Alexa Fluor 488 phalloidin | Thermo Fisher Scientific | Cat# A12379 | IF (1:1000) |
| Chemical compound, drug | MitoTracker Green FM | Thermo Fisher Scientific | Cat# M7514 | |
| Chemical compound, drug | CM-H2DCFDA | Thermo Fisher Scientific | Cat# C6827 | |
| Chemical compound, drug | MitoTracker Red CMXRos | Thermo Fisher Scientific | Cat# M7512 | |
| Chemical compound, drug | Pfu Turbo DNA Polymerase | Agilent Technologies | Cat# 600252-52 | |
| Chemical compound, drug | Streptavidin Resin | Agilent Technologies | Cat# 240105–51 | |
| Chemical compound, drug | 1,6-Hexanediol | Sigma-Aldrich | Cat# 8043081000 | |
| Chemical compound, drug | EDTA-free Protease Inhibitor Cocktail | Sigma-Aldrich | Cat# S8830 | |
| Chemical compound, drug | ANTI-FLAG M2 agarose beads | Sigma-Aldrich | Cat# F2426 | |
| Chemical compound, drug | Polybrene | Sigma-Aldrich | Cat# 107689 | |

*Continued on next page*

*Appendix 1—key resources table continued*

| Reagent type (species) or resource | Designation | Source or reference | Identifiers | Additional information |
|---|---|---|---|---|
| Chemical compound, drug | Diamide | Sigma-Aldrich | Cat# D3648 | |
| Chemical compound, drug | Biotin | Sigma-Aldrich | Cat# B4501 | |
| Chemical compound, drug | Vacuolin-1 | Sigma-Aldrich | Cat# V7139 | |
| Chemical compound, drug | Digitonin | Sangon Biotech | Cat# DG1152 | |
| Chemical compound, drug | ATP-γ-S | Abcam | Cat# ab138911 | |
| Chemical compound, drug | p-Nitrobenzyl mesylate (PNBM) | Abcam | Cat# ab138910 | |
| Software, algorithm | GraphPad Prism | GraphPad | RRID:SCR_002798 | |
| Software, algorithm | Image-Pro Plus 6.0 | Image-Pro Plus | RRID:SCR_016879 | |
| Software, algorithm | VENNY2.1 | https://bioinfogp.cnb.csic.es/tools/venny/index.html | | Coverage of the kinome by KA and databases were analyzed by VENNY2.1 |
| Software, algorithm | KinMap | PMID:28056780 | | Kinome Atlas were presented on kinome tree using KinMap The localization and enrichment levels were reflected by colors and sizes of circles |
| Software, algorithm | SMART | http://smart.embl.de/ | | Domain architecture of each kinase is analyzed by SMART |
| Software, algorithm | IUPred2A | https://iupred2a.elte.hu/ | | Intrinsic disorder tendency is predicted by IUPred2A |
| Software, algorithm | PLAAC | http://plaac.wi.mit.edu/ | | Prion-like domains (PrD.like, red) of each kinase are predicted by PLAAC |

