## [Decision Letter]

Thank you for submitting your article "A subcellular map of the human kinome" for consideration by *eLife*. Your article has been reviewed by 2 peer reviewers, and the evaluation has been overseen by Jonathan Cooper as the Senior and Reviewing Editor. The following individual involved in review of your submission has agreed to reveal their identity: Tony Hunter (Reviewer #1).

Summary:

This paper presents the a large survey of the subcellular localizations of over 80% of the human kinome, as determined by appending an epitope or fluorescent tag at the N or C terminus. The results are catalogued in the Kinase Atlas (KA). A subset of the localizations were investigated further, including those of a putative secreted kinase SGK3 and a mitochondrial kinase, MOK1. The KA will be a useful resource, but it has limitations owing to the use of transient over-expression and possible mislocalization caused by the tagging. These limitations need to be more explicitly stated, and some of the conclusions should be moderated.

Essential revisions:

The original reviews are at the end of this email for your consideration. The reviewers have discussed their reviews with the editor and we consider the following revisions to be essential.

1. The main concern is that some localizations in the KA may be artifactual. This has been an issue with previous large scale tagging surveys in other organisms. Over-expression and the position of the tag could affect subcellular localization. This is more of a concern because the position of the tag (at N or C terminus) seems to be somewhat arbitrary, based on availability of the construct in the library. We therefore recommend softening the claims in the paper, including modifying the Abstract to explicitly state that, for some kinases, either over-expression or the epitope tag may alter localization, so the KA may not in all cases represent the localization of the endogenous protein.

2. It is not clear whether subcellular localization was assessed by the co-staining with specific markers or recognizing specific patterns. This should be clarified. Also, the authors should explain how they loaded subcellular fractions for the blots (equalized for ug protein or for cell number).

3. Conclusions regarding MOK localization and function should be modified (reviewer 1, points 2-3), but we are not requiring additional experiments as they are beyond the scope of this paper.

4. More experiments are needed to understand whether SGK3 is secreted (reviewer 1 point 5). The SGK3 in the media could be soluble or in extracellular vesicles (microvesicles or exosomes). SGK3 has a PX domain for binding endosomes, so it may enter MVBs. The authors should perform high-speed centrifugation and confirm whether SGK3 is in the soluble fraction or pellet. In addition, the text should be modified to clarify that, if secreted, release would be through an unconventional secretion pathway.

*Reviewer #1:*

Here, the authors have carried out a systematic survey of the subcellular localizations of a large fraction of the protein kinases in the human kinome, which they have catalogued in a Kinase Atlas (KA). To do this they tagged 464 out of the 538 human protein kinases with Flag, HA, HA-Flag, Myc or GFP at either the N- or C-terminus, and transiently expressed them by transfection in HeLa cells. ~95% of the tagged kinases could be detected, with about half being expressed at high level. Staining for the pertinent tags and comparison with cells stained for organellar marker proteins allowed the authors to assign each protein kinase to one of 10 subcellular compartments: cytosol (C), nucleus (N), plasma membrane (PM), mitochondrion (MI), endoplasmic reticulum (ER), Golgi apparatus (GL), vesicle (V), cytoskeleton (CS), centrosome (CT), and aggresome (AG). Half the kinases were localized to the cytosol, 15% to the nucleus, and 10% to the plasma membrane, and the rest were divided among other compartments, with several kinases having more than one location. Families of kinases tended to have preferred locations; for instance, as expected the tyrosine kinase family was enriched on the plasma membrane, largely due to the large number of RTKs. Most of the assigned mitochondrial kinases were atypical protein kinases, whereas the RGC and PKL subfamilies were ER/Golgi localized. They found about 70% overlap between their KA localizations and other databases, such as COMPARTMENTS and the Human Proteome Atlas. Some of the discrepancies may depend on whether the tags were N- or C-terminal and the nature of the tag. They found 104 kinase localizations that were uniquely annotated by KA, with 7 localized to the plasma membrane, as confirmed by cell fractionation studies. POMK, the protein-O-mannose kinase, was confirmed to be ER localized, as expected. SGK3 was annotated as specific for the Golgi compartment, but could also be detected in conditioned medium. MAP4K3/MEKK3 was found to decorate tubular structures that were shown to overlap with the ER. Several nuclear kinases were unevenly distributed in puncta of varying size, Including TRIB3 and HIPK2, whereas BMP2K formed puncta in the cytoplasm. Treatment with hexanediol disrupted puncta formed by several of these protein kinases but not puncta formed by other protein kinases, suggesting that the hexanediol-sensitive puncta might form as a result of liquid-liquid phase separation (LLPS). Deletion of predicted IDRs in TRIB3 and BMP2K eliminated their puncta formation. To determine if endogenous BMP2K formed LLPS puncta, they generated BMP2K-HA tag knock-in HeLa cells, and observed BMP2K present in cytoplasmic and/or nuclear puncta, which were abolished by hexanediol treatment. They also found four previously unidentified mitochondrial kinases not annotated in MitoCarta2: MOK, LIMK2, PKN3 and the TNK1 tyrosine kinase. They confirmed the mitochondrial localization of MOK by generating a MOK-Flag knock-in HeLa cell line, and showing that both a full-length 50 kDa form and shorter 20 kDa form were enriched in a mitochondrial fraction. Fractionation and IF analysis indicated that MOK and TNK1 were localized to the mitochondrial intermembrane space, whereas LIMK2 was on the outer mitochondrial membrane. They mapped two short regions in MOK (aa53-103 and aa301-330), required for mitochondrial import. To explore a possible mitochondrial function of MOK, the authors generated MOK knockout A375 melanoma cells and stable MOK knockdown Caki-1 cells, and found that MOK KO/KD cell mitochondria had fewer cristae based on EM analysis, and that both respiration rate and ATP level were reduced, and could be rescued by expression of the MOK2 splice variant but less well by full length MOK1. They showed that the MOK2 splice variant, missing 30 residues near the N terminus of the catalytic domain, lacked in vitro kinase activity, and yet was able to rescue the mitochondrial phenotype of the MOK KO A375 cells. Finally, in investigating possible mechanisms of MOK mitochondrial import, they showed that depletion HSPA9, which is responsible for inward translocation of matrix proteins downstream of the TIM23 complex, reduced MOK import, and that MOK was associated with the IMM protein ATAD3A, which spans between the IMM and OMM, and the SLC25A13 and SLC25A11 solute carriers.

The authors' efforts to carry out a nearly complete (464/538) kinome-wide subcellular localization assessment of the human protein kinases has led to some new findings with regard to protein kinase localization and function, and the resultant Kinome Atlas should be a useful resource for people working in the phosphorylation field. However, this analysis has largely relied on transient expression of tagged protein kinases, and the resultant overexpression and/or the added tag may have led to artifactual subcellular localization, meaning that some fraction of the reported kinase localizations may not be correct. Although many of the analyzed protein kinases appear to have the expected subcellular localization, only the new localizations of two protein kinases, BMPK2 and MOK, were validated at the endogenous level by knocking in a tag, and very few of the other protein kinase localizations were checked by staining for the endogenous proteins. The observation that some protein kinases form hexanediol-sensitive liquid-liquid phase separated (LLPS) puncta could imply previously unsuspected new regulatory biology, but only BMPK2 was shown to form puncta at endogenous levels, and it is possible that transient overexpression might have driven LLPS formation for other protein kinases. In consequence, it would be reassuring to check whether the endogenous protein kinases in question can form hexanediol-sensitive puncta. The new finding that the MOK protein kinase localizes to the mitochondrial intermembrane space is interesting, although more needs to be done to understand how MOK is targeted to this location and how it functions to regulate mitochondrial morphology and activity in a kinase-independent manner.

1. It is unclear to what extent the observed protein kinase localizations require intrinsic protein kinase activity (apart from MOK). This could be checked by expressing kinase-dead mutants of a subset of the protein kinases with interesting localizations. Such studies may be beyond the scope of this paper, but this issue should be discussed.

2. We do not learn anything (new) about subcellular localization signals for protein kinases that have exclusive localizations; for instance, were any new localization motifs found by sequence analysis. Even in the case of MOK, where two short motifs required for mitochondrial import were mapped, it remains unclear how these MOK motifs trigger mitochondrial import, despite the significant number of knockdown and IP/MS experiments carried out to define a possible mechanism of mitochondrial import.

3. MOK was the single understudied protein kinase they examined in depth, and the data demonstrating MOK localization to the mitochondrial intermembrane space in A375 melanoma cells seem quite convincing. However, MOK is expressed at very low levels in most tissues (except testes), and since MOK mitochondrial function was shown to be kinase-independent, its mechanism of action in regulating cristae morphology remains unclear. Also, the effects of MOK KO/KD on mitochondrial function in cells other than A375 cells were not so striking, casting some doubt on whether MOK is likely to be of general importance MOK in mitochondrial morphology and function.

4. Based on Figure 1D, it appears that the authors did not co-stain for a protein kinase of interest and a compartment specific marker to localize the protein kinase, but rather relied on patterns of kinase localization similar to that of an organelle marker protein. If this was what was done, it needs to be explained.

5. Figure 3: Despite these data, it is not clear that SGK3 is secreted in the usual sense of the word, since lacking a signal peptide, one would not expect SGK3 to enter the conventional secretory pathway. For this reason, the authors need to establish whether the SGK3 protein that they identified in the medium is soluble or rather is in or associated with membrane vesicles of some sort, which seems more likely. The authors provide no information in the Materials and methods on how they immunoprecipitated Flag-SGK3 from the culture medium. For instance, was detergent used, in which case SGK3 would have been released from vesicles. The authors could test SGK3 association with vesicles by centrifuging the medium at 100,000 g, which would sediment vesicles, and see whether or not the SGK3 protein remains n the supernatant. Also, as is the case for the other fractionation experiments (see point 6 below), an aliquot of medium corresponding to the fraction of the cell lysate that was analyzed in parallel should be immunoprecipitated to provide the reader a sense of what percentage of the SGK3 population is "secreted". In this regard, it is clear from the figure that the efficiency of SGK3 "secretion" was a lot lower than that of FAM20C, which is a well authenticated secreted protein kinase.

6. For the cell fractionation experiments shown in Figures 3 and 5, and elsewhere, it needs to be made clear in the Materials and methods whether the different fractions represent samples from the same number of cells or rather the same amount of protein. For a fair comparison, the proper way to do this is to normalize to the same number of cells.

7. Figure 4A: Are the protein kinases with prominent IF signals at the nuclear periphery localized inside or outside the nucleus?

8. Figure 5E: The MOK1 and MOK2 alternatively spliced isoforms should generate two proteins separated by ~3 kDa. However, only a single MOK band from the MOK-KI cells is arrowed here and in Figure S5H, and yet one would have expected two bands if MOK2 is expressed in these cells.

9. Figures 6 and 7: It is not clear how the 20 kDa MOK isoform is generated, but since the 3X Flag tag was at the C-terminus, this form must contain the C-terminus of MOK1/2. Can the 20 kDa form be detected in the MOK-KI cells?

10. Figure S7: MOK1 and MOK2 KR mutants were used in panel I, but these mutants are not described in the legend or in the Materials and methods. One assumes that these were designed as kinase-dead mutants -which Lys was mutated to Arg?

*Reviewer #3:*

In this manuscript, Zhang et al. describe the Kinome Atlas (KA), a map of the subcellular localization of 456 kinases representing 85% of the human kinome. The KA is based on experimentally acquired imaging data; the authors have assessed the localization of these kinases using immunofluorescence of overexpressed tagged cDNAs. Importantly, the authors define 10 subcellular compartments that are used to catalog each kinase based on their localization.

Using KA, the authors define the subcellular localization of a collection of kinases that had not been previously studied. In addition, they further report a particular subcellular pattern in some kinases, which the authors attribute to molecular condensates.

Among the novel kinases identified to localize in mitochondria, the authors decided to further characterize MOK. This kinase localized in the intermembrane space, due to the presence of two localization peptides and through a TOM20/TIM23-dependent mechanism. Moreover, knocking out MOK in cells resulted in decreased mitochondrial cristae, reduced oxygen consumption and ATP levels, and increased reactive oxygen species. The authors conclude that MOK plays an important role in mitochondrial function.

Overall, this work is an important resource for many scientists interested in kinase biology because it represents a unique catalog assessing the subcellular localization of virtually all the kinome. The conclusions are well-supported by the data and the authors discover novel insights of poorly characterized kinases. Moreover, it is likely the KA will catalyze novel discoveries based on the unexpected subcellular localization of many kinases.

These are some of the concerns that I have found that require some additional clarification:

1. Although the authors claim that KA has been obtained by a unified experimental approach, this does not seem to be the case. There is a significant number of cDNAs that have been expressed with either N or C-terminal tags, with different affinity tags, or are from different species. A more systematic approach to curate, tag, and express these kinases would yield a much more robust dataset. Along these lines, it would have been extraordinary to have curated the whole human kinome.

2. When looking at the different kinases used in this work and their tags (Sup Table 1), it appears that the authors did not take into consideration the biology of each kinase when deciding whether to use an N- or C-terminal tag. Instead, it was decided based on the availability of Gateway entry clones (which some contain Stop codon, while others don't). For instance, several RTK are tagged at the N-terminal side (KIT, FGFR4, IGFR1, Tie2, and others), which is known to affect their localization, cleavage into the mature form, and interaction to ligands. This limitation has been addressed in some specific cases, where the authors have performed both N- and C-terminal tagging. However, in my opinion, because it is not possible to anticipate how tagging would affect the localization or biology of many kinases that contain regulatory motifs at the N/C terminal regions, the results from KA have to be interpreted with caution, since in some cases could be artifactual.

3. While the authors have not really elucidated the mechanism and/or function of MOK in mitochondria, I believe that it is beyond the scope of this manuscript.

4. Given how valuable these reagents could be for the scientific community, I would recommend that the library is deposited in a public repository (i.e. Addgene).

5. I would recommend the authors to carefully review the list of kinases and their domain architecture and assess whether some of these should be repeated using a different tagging approach. If possible, including some of the protein kinases that are missing would make this resource even more important and useful to the scientific community.

[Editors' note: further revisions were suggested prior to acceptance, as described below.]

Thank you for submitting your article "A subcellular map of the human kinome" for consideration by *eLife*. I have read over your revised article and agree that you have addressed essentially all of the reviewers' concerns, including adding more experiments that were not requested. The paper is much improved and I am ready to accept it.

I noticed a couple of points that you may wish to address before acceptance.

First, regarding MAP4K3, you say the localization is overlapping with microtubules and ER (line 277, Figure 3F). This is not at all obvious from 3F, although the super resolution image in 3G shows excellent co-localization with Sec61. I wonder whether increasing the intensity of the red channel in the merge, and zooming in on region of apparent co-localization, would be helpful.

Second, regarding MOK, line 410, I think you mean that 0.5% digitonin does not permeabilize the IMM. Is there a citation to digitonin permeabilization, and what is the purpose of 0.1% Triton in the figure (it must permeabilize both membranes to allow antibody access but does not allow MOK to leak out). Please explain.

Third, please double check grammar. I will ask staff to proof read carefully upon acceptance, but it will help if you make corrections yourself.

Congratulations on the research. This paper will be a valuable resource for cell biology.

---

## [Author Response]

Essential revisions:The original reviews are at the end of this email for your consideration. The reviewers have discussed their reviews with the editor and we consider the following revisions to be essential.1. The main concern is that some localizations in the KA may be artifactual. This has been an issue with previous large scale tagging surveys in other organisms. Over-expression and the position of the tag could affect subcellular localization. This is more of a concern because the position of the tag (at N or C terminus) seems to be somewhat arbitrary, based on availability of the construct in the library. We therefore recommend softening the claims in the paper, including modifying the Abstract to explicitly state that, for some kinases, either over-expression or the epitope tag may alter localization, so the KA may not in all cases represent the localization of the endogenous protein.

We thank the editor and reviewers for pointing out limitations that should be further explained to readers. We revised the manuscript accordingly to explain potential limitations caused by overexpression and epitope tagging in the abstract, introduction, and conclusion sections (page 2, line 46; page 4, line 102; page17, line 520).

2. It is not clear whether subcellular localization was assessed by the co-staining with specific markers or recognizing specific patterns. This should be clarified. Also, the authors should explain how they loaded subcellular fractions for the blots (equalized for ug protein or for cell number).

In the original manuscript, nucleus was contained by DAPI and other organelles were recognized by specific patterns (revised manuscript page 5 line138). Also in the original manuscript, mitochondrial localizations were confirmed by co-staining with TOM20. We did not co-stain all markers for all kinases in the initial screen due to a tremendous amount of work. However, during the revision, we validated specific ER, Golgi, cytoskeleton and centrosome localizations with a score ≥ 5 by co-staining with respective markers (page 7, line 208). A total of 40 kinases were validated this way, and 36 were consistent with initial conclusions. Initial observations for 4 kinases were not supported by co-staining. TYRO3 previously annotated as mainly GL was in vesicles around GL. ROR1 previously annotated as GL was mostly ER. PAK4 previously annotated as ER was mostly in cytoplasm and in aggresomes. PRKCH previously annotated as ER were found in both MI and ER, and slightly more in MI, suggesting a role in ER-MI interaction. These revisions were documented in Supplementary Table 2, related images were deposited in CIL or in Figure 5C (PRKCH). Plasma membrane localizations were either consistent with previous reports or were validated by fractionation in the original manuscript, thus were not further validated by co-staining. Intracellular vesicles comprising many organelles with distinct markers, and were morphologically obvious to distinguish, thus were not further validated by marker co-staining. We revised workflow of KA mapping (Figure 1—figure supplement 1C).

For western blots of subcellular fractions, the loading was normalized to the same number of cells. We revised respective figure legends for better understanding. In revised Figure 3E and newly added Figure 3—figure supplement 1D for SGK3, as explained in the figure legend, IP and extracellular vesicle fractions were loaded at 5 x of input for better visualization of low-level EV proteins.

3. Conclusions regarding MOK localization and function should be modified (reviewer 1, points 2-3), but we are not requiring additional experiments as they are beyond the scope of this paper.

In response to reviewer 1 point 2, we performed sequence analysis. However, no new linear motif of mitochondrial localization was found. We add this discussion in page 14, line 433; page 17, line 545, and in page 18, line 576.

In response to reviewer 1, point 3, we searched for MOK expression levels. MOK mRNA expression is high in testes in both human and mice (Author response image 1), suggesting a physiological role of MOK in spermatogenesis, during which mitochondria is dramatically remodeled for efficient energy production to support sperm movements. The potential function of mitochondrial MOK in spermatogenesis could be further examined by future investigations. Furthermore, we found that the mRNA level of MOK was deregulated in some cancers in the TCGA database (Author response image 2). For instance, MOK was elevated in cholangiocarcinoma (CHOL), and has highest level in skin cutaneous melanoma (SKCM), suggesting a functional role in cancer. In melanoma cell line A375, knockout of MOK strongly decreases mitochondria cristae number, mitochondrial respiration, and cellular ATP level (Figure 7A-E). In Caki-1 cells, MOK was knocked down by shRNAs, while both shRNAs also reduces mitochondria cristae number (p < 0.0001), reduction of cellular ATP level was only marginally inhibited by shRNA #2 (p = 0.0377) (Figure 7-supplement 1C-E). This could be due to cell type-dependent metabolic reprogramming which may produce ATP through other pathways, such as glycolysis. We modified statements in page 18, last paragraph.

**Author response image 1. respfig1:** Expression of MOK in human and mouse tissues. Data was analyzed from the HPA RNA-seq normal tissues project (human) and the Mouse ENCODE transcriptome data.

**Author response image 2. respfig2:** Expression of *MOK* in human cancers in TCGA database as analyzed by FireBrowse (http://firebrowse.org/).

4) More experiments are needed to understand whether SGK3 is secreted (reviewer 1 point 5). The SGK3 in the media could be soluble or in extracellular vesicles (microvesicles or exosomes). SGK3 has a PX domain for binding endosomes, so it may enter MVBs. The authors should perform high-speed centrifugation and confirm whether SGK3 is in the soluble fraction or pellet. In addition, the text should be modified to clarify that, if secreted, release would be through an unconventional secretion pathway.

We thank the reviewer for this constructive suggestion. To determine whether SGK3 is secreted by conventional ER/Golgi pathway or MVB pathway, we first examined the effect of brefeldin A (BFA), an inhibitor of ER/Golgi‐trafficking. As shown in the revised Figure 3E, BFA blocked secretion of FAM20C, but not SGK3, suggesting it was not secreted through ER/Golgi pathway. Furthermore, we performed ultracentrifugation at 100,000 g, and found SGK3 but not FAM20C in pellet (revised Figure 3—figure supplement 1D). These findings indicate that SGK3 is secreted unconventionally through the MVB pathway. Text was modified accordingly.

Reviewer #1:[…]The authors' efforts to carry out a nearly complete (464/538) kinome-wide subcellular localization assessment of the human protein kinases has led to some new findings with regard to protein kinase localization and function, and the resultant Kinome Atlas should be a useful resource for people working in the phosphorylation field. However, this analysis has largely relied on transient expression of tagged protein kinases, and the resultant overexpression and/or the added tag may have led to artifactual subcellular localization, meaning that some fraction of the reported kinase localizations may not be correct. Although many of the analyzed protein kinases appear to have the expected subcellular localization, only the new localizations of two protein kinases, BMPK2 and MOK, were validated at the endogenous level by knocking in a tag, and very few of the other protein kinase localizations were checked by staining for the endogenous proteins. The observation that some protein kinases form hexanediol-sensitive liquid-liquid phase separated (LLPS) puncta could imply previously unsuspected new regulatory biology, but only BMPK2 was shown to form puncta at endogenous levels, and it is possible that transient overexpression might have driven LLPS formation for other protein kinases. In consequence, it would be reassuring to check whether the endogenous protein kinases in question can form hexanediol-sensitive puncta. The new finding that the MOK protein kinase localizes to the mitochondrial intermembrane space is interesting, although more needs to be done to understand how MOK is targeted to this location and how it functions to regulate mitochondrial morphology and activity in a kinase-independent manner.

We thank the reviewer for a positive opinion on our manuscript. Regarding localization of endogenous protein kinases, we have also tried staining with commercially available anti-endogenous protein antibodies for TRIB3 and PRPF4B, two kinases with LLPS-like puncta. Nevertheless, the specificity of these antibodies could not be satisfactorily verified. These data were thus not included.

1. It is unclear to what extent the observed protein kinase localizations require intrinsic protein kinase activity (apart from MOK). This could be checked by expressing kinase-dead mutants of a subset of the protein kinases with interesting localizations. Such studies may be beyond the scope of this paper, but this issue should be discussed.

To determine the effect of protein kinase activity in its localization, we selected 2 PM kinases, 2 MI kinases, 2 kinases with LLPS-like puncta, and MAP4K3 with tubule-like structures. Kinase inactive mutants were made, and localizations were examined. Except HIPK2, all other inactive kinases showed localization patterns similar to their wildtype counterparts (Figure 3—figure supplement 1H). While wildtype HIPK2 were mostly in nuclear puncta, inactive HIPK2 was diffusive in cytoplasm in more than 50% of cells. Thus, case by case, subcellular localizations of protein kinases could be dependent on their activity. This new result is discussed in page 10, line 295.

2. We do not learn anything (new) about subcellular localization signals for protein kinases that have exclusive localizations; for instance, were any new localization motifs found by sequence analysis. Even in the case of MOK, where two short motifs required for mitochondrial import were mapped, it remains unclear how these MOK motifs trigger mitochondrial import, despite the significant number of knockdown and IP/MS experiments carried out to define a possible mechanism of mitochondrial import.

It would be interesting to decode consensus linear motifs from newly identified kinases with specific localizations. In order to do so, we have searched for known motifs using Motif Scan (https://myhits.sib.swiss/cgi-bin/motif_scan/), MOTIFS (https://molbiol-tools.ca/Motifs.htm), and searched for new linear motifs using MEME Suite (http://meme-suite.org/). To exclude the interference of the kinase domain, which is highly homologous, we have also performed analysis using sequences excluding the kinase domain. However, no new motifs related to mitochondrial or plasma membrane localizations were found. Other localizations were not analyzed. We postulate that unconventional localization signals may be diverse and present at low ratios. There may be a higher chance to identify such motifs when analysis was performed on the proteome level beyond the kinome. In addition, as we discussed in page 17, line 545, novel sorting mechanisms such as signal patches are possible, which could not be predicted from secondary sequence.

3. MOK was the single understudied protein kinase they examined in depth, and the data demonstrating MOK localization to the mitochondrial intermembrane space in A375 melanoma cells seem quite convincing. However, MOK is expressed at very low levels in most tissues (except testes), and since MOK mitochondrial function was shown to be kinase-independent, its mechanism of action in regulating cristae morphology remains unclear. Also, the effects of MOK KO/KD on mitochondrial function in cells other than A375 cells were not so striking, casting some doubt on whether MOK is likely to be of general importance MOK in mitochondrial morphology and function.

As the reviewer pointed out, MOK expression is high in testes in both human and mice (Author response image 1), suggesting a physiological role of MOK in spermatogenesis, during which mitochondria is dramatically remodeled for efficient energy production to support sperm movements. The potential function of mitochondrial MOK in spermatogenesis could be further examined by future investigations. Furthermore, we found that the mRNA level of MOK was deregulated in some cancers in the TCGA database (Author response image 2). For instance, MOK was elevated in cholangiocarcinoma (CHOL), and has highest level in skin cutaneous melanoma (SKCM), suggesting a functional role in cancer. In melanoma cell line A375, knockout of MOK strongly decreases mitochondria cristae number, mitochondrial respiration, and cellular ATP level (Figure 7A-E). In Caki-1 cells, MOK was knocked down by shRNAs, while both shRNAs also reduces mitochondria cristae number (p < 0.0001), reduction of cellular ATP level was only marginally inhibited by shRNA #2 (p = 0.0377) (Figure 7—figure supplement 1C-E). This could be due to cell type-dependent metabolic reprogramming which may produce ATP through other pathways, such as glycolysis.

4. Based on Figure 1D, it appears that the authors did not co-stain for a protein kinase of interest and a compartment specific marker to localize the protein kinase, but rather relied on patterns of kinase localization similar to that of an organelle marker protein. If this was what was done, it needs to be explained.

In the original manuscript, nucleus was contained by DAPI and other organelles were recognized by specific patterns (revised manuscript page 5 line138). Also in the original manuscript, mitochondrial localizations were confirmed by co-staining with TOM20. We did not co-stain all markers for all kinases in the initial screen due to a tremendous amount of work. However, during the revision, we validated specific ER, Golgi, cytoskeleton and centrosome localizations with a score ≥ 5 by co-staining with respective markers (page 7, line 208). A total of 40 kinases were validated this way, and 36 were consistent with initial conclusions. Initial observations for 4 kinases were not supported by co-staining. TYRO3 previously annotated as mainly GL was in vesicles around GL. ROR1 previously annotated as GL was mostly ER. PAK4 previously annotated as ER was mostly in cytoplasm and in aggresomes. PRKCH previously annotated as ER were found in both MI and ER, and slightly more in MI, suggesting a role in ER-MI interaction. These revisions were documented in Supplementary Table 2, related images were deposited in CIL or in Figure 5C (PRKCH). Plasma membrane localizations were either consistent with previous reports or were validated by fractionation in the original manuscript, thus were not further validated by co-staining. Intracellular vesicles comprising many organelles with distinct markers, and were morphologically obvious to distinguish, thus were not further validated by marker co-staining. We revised workflow of KA mapping (Figure 1—figure supplement 1C).

5. Figure 3: Despite these data, it is not clear that SGK3 is secreted in the usual sense of the word, since lacking a signal peptide, one would not expect SGK3 to enter the conventional secretory pathway. For this reason, the authors need to establish whether the SGK3 protein that they identified in the medium is soluble or rather is in or associated with membrane vesicles of some sort, which seems more likely. The authors provide no information in the Materials and methods on how they immunoprecipitated Flag-SGK3 from the culture medium. For instance, was detergent used, in which case SGK3 would have been released from vesicles. The authors could test SGK3 association with vesicles by centrifuging the medium at 100,000 g, which would sediment vesicles, and see whether or not the SGK3 protein remains n the supernatant. Also, as is the case for the other fractionation experiments (see point 6 below), an aliquot of medium corresponding to the fraction of the cell lysate that was analyzed in parallel should be immunoprecipitated to provide the reader a sense of what percentage of the SGK3 population is "secreted". In this regard, it is clear from the figure that the efficiency of SGK3 "secretion" was a lot lower than that of FAM20C, which is a well authenticated secreted protein kinase.

We thank the reviewer for this constructive suggestion. To immunoprecipitate SGK3 from medium, 1% NP-40 was added before IP. This information was updated in Materials and methods. To determine whether SGK3 is secreted by conventional ER/Golgi pathway or MVB pathway, we first examined the effect of brefeldin A (BFA), an inhibitor of ER/Golgi‐trafficking. As shown in the revised Figure 3E, BFA blocked secretion of FAM20C, but not SGK3, suggesting it was not secreted through ER/Golgi pathway. Furthermore, we performed ultracentrifugation at 100,000 g, and found SGK3 but not FAM20C in pellet (revised Figure 3—figure supplement 1D). These findings indicate that SGK3 is secreted unconventionally through the MVB pathway. Text was modified accordingly. In the revised Figure 3E, loading was normalized by the number of input cells, and IP samples were loaded at 5 × of input. Indeed, the secretion of SGK3 is much lower in comparison to FAM20C.

6. For the cell fractionation experiments shown in Figures 3 and 5, and elsewhere, it needs to be made clear in the Materials and methods whether the different fractions represent samples from the same number of cells or rather the same amount of protein. For a fair comparison, the proper way to do this is to normalize to the same number of cells.

We apologize for not explain explicitly. But indeed, the loading was normalized to the same number of cells. We revised respective figure legends for better understanding.

7. Figure 4A: Are the protein kinases with prominent IF signals at the nuclear periphery localized inside or outside the nucleus?

To answer the reviewer’s question, we stained cells for the kinase and a nuclear envelope marker protein Lamin A/C. Images were captured by Zeiss Airyscan high-resolution microscopy. The results indicate that all 6 kinases were overlapping or within the boundaries of cell nuclei. The new data was provided in revised Figure 4—figure supplement1A.

8. Figure 5E: The MOK1 and MOK2 alternatively spliced isoforms should generate two proteins separated by ~3 kDa. However, only a single MOK band from the MOK-KI cells is arrowed here and in Figure S5H, and yet one would have expected two bands if MOK2 is expressed in these cells.

It is possible that one isoform of MOK is dominating in HeLa cells. We thus PCR amplified MOK and sequenced TA clones for the differential region. Indeed, among 17 clones sequenced, 14 (82%) were MOK1 and 3 were MOK2. Thus, HeLa cells mainly express MOK1, which accounts for a major band in MOK-KI cells. Furthermore, a non-specific band slightly smaller than MOK1 was found in HeLa cells, which also interferes with identifying a minor band of MOK.

9. Figures 6 and 7: It is not clear how the 20 kDa MOK isoform is generated, but since the 3X Flag tag was at the C-terminus, this form must contain the C-terminus of MOK1/2. Can the 20 kDa form be detected in the MOK-KI cells?

Yes, the 20KDa band could be detected in MOK-KI cells by anti-Flag western blotting (Figure. 5E). This band is MOK because MOK siRNA reduced the intensity of this band (Figure 5—figure supplement 1H and I). It could be a different isoform or a degradation product. But the exact mechanism for the generation of this band is unknown.

10. Figure S7: MOK1 and MOK2 KR mutants were used in panel I, but these mutants are not described in the legend or in the Materials and methods. One assumes that these were designed as kinase-dead mutants -which Lys was mutated to Arg?

Yes, KR denotes a kinase inactive mutant generated by mutating the ATP binding Lys to Arg. The figure legend of Figure 7—figure supplement 1I was revised.

Reviewer #3:[…]These are some of the concerns that I have found that require some additional clarification:1. Although the authors claim that KA has been obtained by a unified experimental approach, this does not seem to be the case. There is a significant number of cDNAs that have been expressed with either N or C-terminal tags, with different affinity tags, or are from different species. A more systematic approach to curate, tag, and express these kinases would yield a much more robust dataset. Along these lines, it would have been extraordinary to have curated the whole human kinome.

Indeed, it was our goal to map the whole human kinome. However, this was limited first by resources of our lab, and second by intrinsic characters of some leftover kinases. While many of these kinases were cloned, protein expression was undetectable when transfected. We revised statement of “a unified condition” to “in the same cell line transfected and cultured in a similar condition”.

2. When looking at the different kinases used in this work and their tags (Sup Table 1), it appears that the authors did not take into consideration the biology of each kinase when deciding whether to use an N- or C-terminal tag. Instead, it was decided based on the availability of Gateway entry clones (which some contain Stop codon, while others don't). For instance, several RTK are tagged at the N-terminal side (KIT, FGFR4, IGFR1, Tie2, and others), which is known to affect their localization, cleavage into the mature form, and interaction to ligands. This limitation has been addressed in some specific cases, where the authors have performed both N- and C-terminal tagging. However, in my opinion, because it is not possible to anticipate how tagging would affect the localization or biology of many kinases that contain regulatory motifs at the N/C terminal regions, the results from KA have to be interpreted with caution, since in some cases could be artifactual.

We thank the editor and reviewers for pointing out limitations that should be further explained to readers. We revised the manuscript accordingly to explain potential limitations caused by overexpression and epitope tagging in the abstract, introduction, and conclusion sections (page 2, line 46; page 4, line 102; page17, line 520).

3. While the authors have not really elucidated the mechanism and/or function of MOK in mitochondria, I believe that it is beyond the scope of this manuscript.

We thank the reviewer for understanding. The function and mechanism of MOK in mitochondria will be further investigated in future works.

4. Given how valuable these reagents could be for the scientific community, I would recommend that the library is deposited in a public repository (i.e. Addgene).

Kinase plasmids are currently available for request from the core facility of the Life Sciences Institute of Zhejiang University by emailing to lsicf@zju.edu.cn. We will try to deposit to Addgene once the COVID pandemic is resolved. For easier access to images of KA, images were deposited to the Cell Image Library database during the revision (http://flagella.crbs.ucsd.edu/pages/kinome_atlas?token=dHqMbfi06S). This replaces the original Supplementary document S1. A reviewer link was provided in the revised manuscript, and this will be replaced by a public link if accepted. The current reviewer portal allows browsing of images but is not searchable. However, the public version will be fully searchable for easier use of the dataset.

5. I would recommend the authors to carefully review the list of kinases and their domain architecture and assess whether some of these should be repeated using a different tagging approach. If possible, including some of the protein kinases that are missing would make this resource even more important and useful to the scientific community.

We thank the reviewer for suggestions on further improving this resource. At the library construction stage, we have tried our best to either cloning or inquiring the missing kinases. However, the leftover kinases are mostly large genes, and we failed to clone them due to various reasons. Furthermore, many leftover kinases were undetectable when transfected, although we cloned the cDNA into multiple plasmid vectors. This is likely due to intrinsic features of the kinases. To assess the impact of N-tag, we compared KA to COMPARTMENTS. Among 187 N-tagged kinases with high confidence localizations in COMPARTMENTS, 131 were consistent with that found in KA, indicating that the N-terminal tag did not generally interfere with kinase localization. Among the 56 differential kinases, 34 have membrane-related or mitochondrial localizations in COMPARTMENTS. An additional 4 N-tagged kinases have transmembrane regions and predicted signal peptides. Thus, we constructed C-tagged version of these kinases, and imaging was done. KA was then revised according to the results (Supplementary Table 2). This procedure is part of KA and was described in page 7, line 204. This workflow was described in Figure 1—figure supplement 1C.

[Editors' note: further revisions were suggested prior to acceptance, as described below.]

I noticed a couple of points that you may wish to address before acceptance.First, regarding MAP4K3, you say the localization is overlapping with microtubules and ER (line 277, Figure 3F). This is not at all obvious from 3F, although the super resolution image in 3G shows excellent co-localization with Sec61. I wonder whether increasing the intensity of the red channel in the merge, and zooming in on region of apparent co-localization, would be helpful.Second, regarding MOK, line 410, I think you mean that 0.5% digitonin does not permeabilize the IMM. Is there a citation to digitonin permeabilization, and what is the purpose of 0.1% Triton in the figure (it must permeabilize both membranes to allow antibody access but does not allow MOK to leak out). Please explain.Third, please double check grammar. I will ask staff to proof read carefully upon acceptance, but it will help if you make corrections yourself.

We have revised the manuscript according to your advice. Figure 3F was revised so that insets of enlarged views were included to make colocalizations more visible. Figure legend was revised accordingly. The text for results of Figure 6B, line 410 was revised to explain the difference between Triton X-100 and digitonin with a reference. In addition, the manuscript was proofread for grammars again with mistakes corrected.